# 3D crustal stress state of Germany according to a data-calibrated geomechanical model

Steffen Ahlers[1], Andreas Henk[1], Tobias Hergert[1], Karsten Reiter[1], Birgit Müller[2], Luisa Röckel[2], Oliver Heidbach[3], Sophia Morawietz[3], Magdalena Scheck-Wenderoth[3], Denis Anikiev[3]

[1]Institut für Angewandte Geowissenschaften, TU Darmstadt, 64287 Darmstadt, Germany
[2]Institut für Angewandte Geowissenschaften, KIT, 76131 Karlsruhe, Germany
[3]Deutsches GeoForschungsZentrum (GFZ), 14473 Potsdam, Germany

*Correspondence to*: Steffen Ahlers (ahlers@geo.tu-darmstadt.de)

**Abstract.** The contemporary stress state in the upper crust is of great interest for geotechnical applications and basic research likewise. However, our knowledge of the crustal stress field from the data perspective is limited. For Germany basically two datasets are available: Orientations of the maximum horizontal stress ($S_{Hmax}$) and the stress regime as part of the World Stress Map (WSM) database as well as a complementary compilation of stress magnitude data of Germany and adjacent regions. However, these datasets only provide pointwise, incomplete and heterogeneous information of the 3D stress tensor. Here, we present a geomechanical-numerical model that provides a continuous description of the contemporary 3D crustal stress state on a regional scale for Germany. The model covers an area of about 1000 x 1250 km$^2$ and extends to a depth of 100 km containing seven units, with specific material properties (density and elastic rock properties) and laterally varying thicknesses: A sedimentary unit, four different units of the upper crust, the lower crust and the lithospheric mantle. The model is calibrated by the two datasets to achieve a best-fit regarding the $S_{Hmax}$ orientations and the minimum horizontal stress magnitudes ($S_{hmin}$). The modelled orientations of $S_{Hmax}$ are almost entirely within the uncertainties of the WSM data used and the $S_{hmin}$ magnitudes fit to various datasets well. Only the $S_{Hmax}$ magnitudes show locally significant deviations, primarily indicating too low values in the lower part of the model. The model is open for further refinements regarding model geometry, e.g., additional layers with laterally varying material properties, and incorporation of future stress measurements. In addition, it can provide the initial stress state for local geomechanical models with a higher resolution.

## 1 Introduction

Knowledge about the stress state in the upper crust is of great importance for many economic and scientific questions. For example, wellbore stability (Bell, 2003; Kristiansen, 2004), operation and stimulation of hydrocarbon and geothermal reservoirs (Altmann et al., 2014; Azzola et al., 2019; Henk, 2009; Smart et al., 2014), slip and dilation tendency of existing faults and fractures (Hettema, 2020; Konstantinovskaya et al., 2012), underground mining (Brady and Brown, 2004) and deep tunneling (Diederichs et al., 2004). Furthermore, it plays a decisive role in the search for a disposal site for high-level radioactive waste, since it is crucial for the short and long-term safety of a possible repository. (StandAG, 2017; nagra, 2008; BGR, 2015). For all these applications the contemporary stress state is a key parameter and thus, the quantification of the complete 3D stress tensor is essential.

However, from the data perspective, our knowledge of the stress state in Western Central Europe is limited in particular regarding stress magnitudes information. Public stress information is provided by the World Stress Map (WSM) project which supplies a global database of the orientation of maximum horizontal stress ($S_{Hmax}$) and the stress regime (Heidbach et al., 2018) and by a compilation of stress magnitude data for Germany and adjacent regions of Morawietz et al. (2020). However, these two datasets contain only pointwise information, which is incomplete as only a subset of the stress tensor components is provided and their spatial distribution is sparse and irregular (Fig. 1a).

To provide a continuous description of the 3D stress tensor in the upper crust on a regional scale, we developed the first 3D

geomechanical model covering Germany (Fig. 1). Our model comprises seven units, with specific material properties and laterally varying thicknesses: A sedimentary unit, four different units of the upper crust, the lower crust and the lithospheric mantle. The finite element method (FEM) is used to solve the partial differential equation which describes the equilibrium of body and surface forces within an inhomogeneous medium. Our input parameters are density and elastic material properties (Young's modulus and Poisson's ratio). The model is calibrated using appropriate initial conditions and displacement

boundary conditions to find a best-fit with respect to the stress orientation and magnitude datasets described above. This modelling approach has been used for a wide range of scales and different tectonic settings (Buchmann and Connolly, 2007; Heidbach et al., 2014; Hergert and Heidbach, 2011; Hergert et al., 2015; Reiter and Heidbach, 2014).

## 2 Fundamentals and state of the art

### 2.1 Geology and tectonic setting of the study area

The crustal and lithospheric structure in the model domain reflects the complex geodynamic evolution of Central Europe since Precambrian times (McCann, 2008; Meschede and Warr, 2019) (Fig. 1c and d). The north-eastern part of the study area belongs to the cratonic unit of Baltica and, more specifically, the East European Craton (EEC). This unit consists mainly of high-grade magmatic and metamorphic rocks of Precambrian and early Palaeozoic age. Crustal thickness in the area is about 50 km and a thick mantle lithosphere down to depths of 200 km has been observed (Mazur et al., 2015). The EEC is

separated from the Avalonia microplate to the south-west by the Tornquist Suture and the Thor Suture, respectively (e.g Linnemann et al., 2008 and various references therein). At this boundary, a sharp transition to the significantly thinner crustal and lithospheric thicknesses typical for Paleozoic and Mesozoic Europe can be observed (Ziegler and Dèzes, 2006). Western Baltica and eastern Avalonia got into contact during closure of the Tornquist Ocean during Ordovician to Silurian times and the Caledonian orogeny, respectively. At this stage, Laurussia (composed of Laurentia, Baltica and Avalonia) was

formed, whose continental crust makes up the northern and eastern part of the study area.

The central part of the model domain comprises the southern part of Avalonia as well as Amorica – microplates and terrane assemblages - which collided during the Variscan orogeny in Late Palaeozoic times (Franke, 1989, 2006; Meschede and Warr, 2019). The low-grade metamorphic rocks of the Rhenohercynian Zone (sensu Kossmat, 1927) represent passive margin sediments which were deposited on thinned crust of south-eastern Avalonia. South-eastward directed subduction and

closure of the Rheic Ocean led to the formation of an active margin at the northern rim of Amorica, which nowadays comprises the medium-grade metamorphic and magmatic rocks of the so-called Mid German Crystalline High (Oncken, 1997). Further to the south, the Saxothuringian and Moldanubian Zone represent the remnants of the internal zone of the Variscan orogen with medium- to high-grade metamorphic rocks and abundant granitoids presently exposed at surface. Crustal thickness in this part of the model domain and outside the areas affected by Cenozoic rifting and mountain building

is typically in the order of 30 km (Ziegler and Dèzes, 2006). The late Cretaceous to Paleogene evolution was influenced by NE-directed Africa-Iberia-Europe convergence which led to intraplate contraction and inversion of NW-SE striking structural elements (Kley and Voigt, 2008). The final stage of this phase coincides with W-E to NW-SE directed extension and the onset of rifting in the Upper Rhine Graben (URG) and Eger Graben, among others (Kley et al., 2008).

The southernmost parts of the study area are located in the so-called ALCAPA (Alps–Carpathians–Pannonian) unit or

terrane (e.g., Brückl et al., 2010; Schmid et al., 2004). Its geodynamic evolution is closely related to the collision between Europe and the Adriatic-Apulian microplate leading to the Alpine orogen. Since Eocene times, its northern foreland is characterized by N-S to NW-SE directed compression and thrusting, respectively (Reicherter et al., 2008).

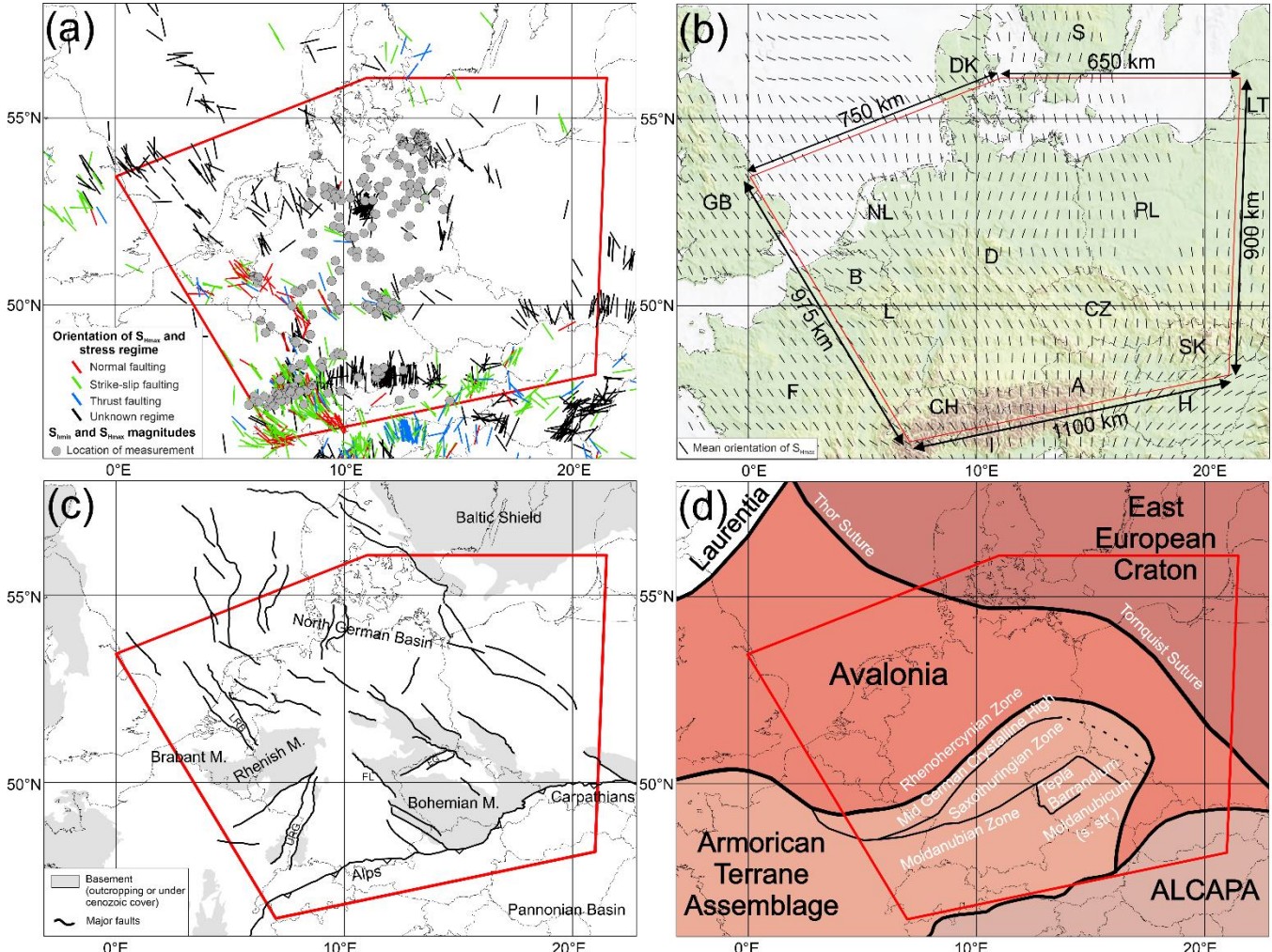

**Figure 1: Maps of Western Central Europe, the red polygon indicates the model area. (a) Overview of calibration data used. Color-coded lines indicate the orientation of S$_{Hmax}$ and the stress regime of the WSM (Heidbach et al., 2018) and additional data of Levi et al. (2019). Grey dots show the positions of stress magnitude data of Morawietz et al. (2020). (b) Topography and mean S$_{Hmax}$ orientations on a regular 0.5° grid derived from the WSM. Each grid point requires at least ten data points within a fixed search radius of 200 km (details in Sect. 4.1). The topography is based on Smith and Sandwell (1997). (c) Tectonic framework of the model area based on Asch (2005) and Kley and Voigt (2008). EG - Eger Graben, FL - Franconian Line, LRB - Lower Rhine Basin, M. - Massif, URG - Upper Rhine Graben. (d) Overview of the crustal units in Central Europe (modified after Kroner et al., 2010 and Brückl et al., 2010). Black titles show tectonic units and white titles sutures and Variscan units.**

## 2.2 Basics of the crustal stress state

The stress state at a given point can be described by a second rank tensor (Fig. 2a) with Pascal (1 Pa = N m$^{-2}$) as the basic unit. Due to its symmetry properties, only six out of nine components are independent from each other (e.g. Jaeger et al., 2011). In the principal axis system, the off-diagonal components vanish and the remaining three components are the principal stresses $\sigma_1$, $\sigma_2$ and $\sigma_3$. Their orientations and magnitudes describe the absolute stress state (Fig. 2b). Assuming that the vertical stress (S$_V$) is one of the three principal stresses (Fig. 2c), the orientation of this so-called reduced stress tensor is determined by the orientation of S$_{Hmax}$. Given that S$_V$ can be approximated by depth and density of the overburden, the remaining unknowns are the magnitudes of S$_{Hmax}$ and the minimum horizontal stress (S$_{hmin}$).

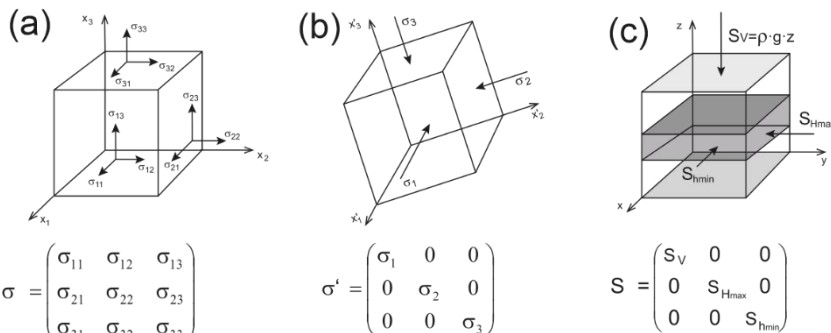

**Figure 2: (a) The nine components of the stress tensor define the stress state at an arbitrary point and enable to compute the stress vector on any surface through that point. To describe the stress tensor components an infinitely small cube with uniform surfaces is used. (b) Due to the conservation of momentum, the stress tensor is symmetric and thus a coordinate system exists where shear stresses vanish along the faces of the cube. In this principal axis system, the remaining three stresses are the principal stresses. (c) Assuming that the vertical stress (the overburden) in the Earth crust $S_V = g \cdot \rho \cdot z$ is a principal stress (g is gravitational acceleration, $\rho$ is the rock density, z is depth), the two horizontal stresses $S_{hmin}$ and $S_{Hmax}$, the minimum and maximum horizontal stress, respectively, are principal stresses as well. This so-called reduced stress tensor is fully determined with four components: The $S_{Hmax}$ orientation and the magnitudes of $S_V$, $S_{hmin}$ and $S_{Hmax}$. (Heidbach et al., 2018)**

The stress state of the continental crust is influenced by stress sources on different scales from several meters up to several thousand kilometers. First-order stress sources (>100 km) related to plate boundary forces, e.g. ridge push or slab pull, second-order stress sources (~100 km) related to large volume forces, e.g. lithospheric flexure due to mountain ranges or deglaciation and third-order stress sources (< 100 km) related to local density or stiffness contrasts, e.g. faults or diapirs. Second and third-order stress sources are able to disturb the overall stress orientation trend from regional through local to reservoir scale. (Heidbach et al., 2007)

**2.3 Data compilation of stress tensor components**

The orientation of the stress tensor in the Earth's crust is provided by the WSM database, which is a global compilation providing data on the $S_{Hmax}$ orientation and the stress regime (Heidbach et al., 2018). This stress information is derived from a variety of methods, primarily earthquake focal mechanism solutions, borehole breakouts and drilling-induced tensile fractures (from borehole image or multi-arm caliper log data), in-situ stress measurements (overcoring or hydraulic fracturing) and geologic indicators, such as fault slip and volcanic vent alignment (Amadei and Stephansson, 1997; Ljunggren et al., 2003; Schmitt et al., 2012). The stress information in the WSM database is compiled in a standardized format and quality-ranked for reliability and comparability on a global scale (Heidbach et al., 2010; Zoback, 1992). For Germany and adjacent regions, the $S_{Hmax}$ orientations have been re-evaluated recently (Reiter et al., 2016; Reiter et al., 2015). The new data have been integrated in the latest WSM database release (Heidbach et al., 2018). For stress magnitude data, Morawietz et al. (2020) published a publicly accessible database with 568 data records including a quality assessment of the data for Germany and adjacent regions. These two datasets (Fig. 1a), the $S_{Hmax}$ orientation of the WSM with some additional data of Levi et al. (2019) and the stress magnitude database are used to calibrate the geomechanical model.

**2.4 Previous models**

Modelling the contemporary crustal stress in Western Central Europe has been addressed by various authors since the mid-1980s. However, except the model of Buchmann and Connolly (2007) which provide a 3D model of the broader URG, all models are 2D and with a strong emphasis on the $S_{Hmax}$ orientation and little regarding the stress magnitudes. Table 1 gives a short overview of their key technical characteristics. If several model versions are published by one author, the most current one is listed. In general, different plastic and elastic material properties have been tested so far and also various boundary conditions have been applied. For a detailed overview we refer to e.g. Cacace (2008), Heidbach et al. (2007) or Jarosiński et al. (2006).

The results indicate different main factors influencing the contemporary stress state. The majority of the studies have found lateral stiffness contrasts in the lithosphere, such as the Bohemian Massif, the Elbe Fault Zone or the Avalonia-EEC boundary (Cacace, 2008; Grünthal and Stromeyer, 1994; Jarosiński et al., 2006; Marotta et al., 2002) and isostatic effects
(Bada et al., 2001; Kaiser et al., 2005; Jarosiński et al., 2006) to be the main cause of stress perturbations. In addition, faults or fault zones are held responsible for third or second-order stresses, respectively (Jarosiński et al., 2006; Kaiser et al., 2005). An indirect influence of the depth position of the asthenosphere lithosphere boundary (LAB) and the resulting temperature contrasts and changed mechanical properties are described by Cacace (2008).

**Table 1: Overview of regional scale stress models within the model area. If several model versions are published, the most current one is listed. The (X) is used for Buchmann and Connolly (2007), since the boundary conditions are not derived from the plate boundary forces, but still represent them.**

| Author | Model area | Model dimension | Material law | | Boundary and initial conditions | | Variations | | | |
|---|---|---|---|---|---|---|---|---|---|---|
| | | | elastic | plastic | plate boundary forces | gravity induced forces | density | friction coefficient (faults) | thickness | plastic or elastic material parameters |
| Andeweg, 2002 | European part of Eurasia | 2D | X | | | X | X | | X | |
| Bada et al., 2001 | Pannonian Basin and surrounding orogens | 2D | X | | X | | | | | X |
| Buchmann and Connolly, 2007 | Upper Rhine Graben | 3D | X | X | (x) | | X | | | X |
| Cacace, 2008 | Central European Basin System | 2D, thin sheet | | X | X | | | | | X |
| Goelke and Coblentz, 1996 | European part of Eurasia | 2D | X | | X | | X | | | |
| Grünthal and Stromeyer, 1994 | East and Central Eurasia | 2D | X | | X | | | | | X |
| Jarosiński et al., 2006 | East and Central Europe | 2D | X | | | X | | X | | X |
| Kaiser et al., 2005 | North Germany and South Scandinavia | 2D, thin plate | | X | X | X | | X | | X |
| Marotta et al., 2002 | Germany | 2D, thin sheet | | X | X | | | | X | X |
| Warners-Ruckstuhl et al., 2013 | Eurasia | 2D, thin shell | X | | X | | X | X | X | |

## 3. Model setup

### 3.1 Conceptual modelling approach

To model the contemporary 3D stress field of the upper crust we assume linear elasticity and neglect thermal stresses and pore pressure effects. With these assumptions, the partial differential equation of the equilibrium of forces has to be solved (Jaeger et al., 2007). The two contributing forces are volume forces from gravitational acceleration and surfaces forces that are mainly attributed to plate tectonics. The latter are key drivers for the tectonic stress that we observe and they are parametrized with displacement boundary conditions that are chosen in a way that the resulting stresses deliver a best-fit
with respect to the model-independent stress data. Although this displacement boundary conditions are mainly representing the tectonic stresses they are not derived from these. Accordingly, our results do not allow any conclusions regarding the sources of crustal stress in the model area. This process is called model calibration which can also be used to estimate model uncertainties by means of standard deviation (Ziegler et al., 2016; Ziegler and Heidbach, 2020). The technical procedure is

presented in Fig. 3 with a schematic general workflow. The individual text boxes are color-coded indicating the four major

steps.

The model geometry reflects the contemporary distribution of rock properties such as density and stiffness and Poisson's ratio. An appropriate initial stress equilibrates the gravitational stresses and resembles a reference stress state (Fischer and Henk, 2013; Hergert et al., 2015; Reiter and Heidbach, 2014). The orientations of the lateral model boundaries where the displacement boundary conditions are applied are chosen in such a way that the mean $S_{Hmax}$ orientation (Fig 1b) is

perpendicular or parallel to them.

For the solution of the partial differential equation of the equilibrium of forces, we use the FEM to estimate an approximated numerical solution. The FEM is appropriate as it allows discretizing complex geometries with unstructured meshes. The commercial FEM software package Abaqus™ v2019 is used. For post-processing we are using Tecplot 360™ enhanced with the GeoStress add-on (Stromeyer and Heidbach, 2017). For the construction and discretization of the 3D model geometry

GOCAD™ and HyperMesh™ are used.

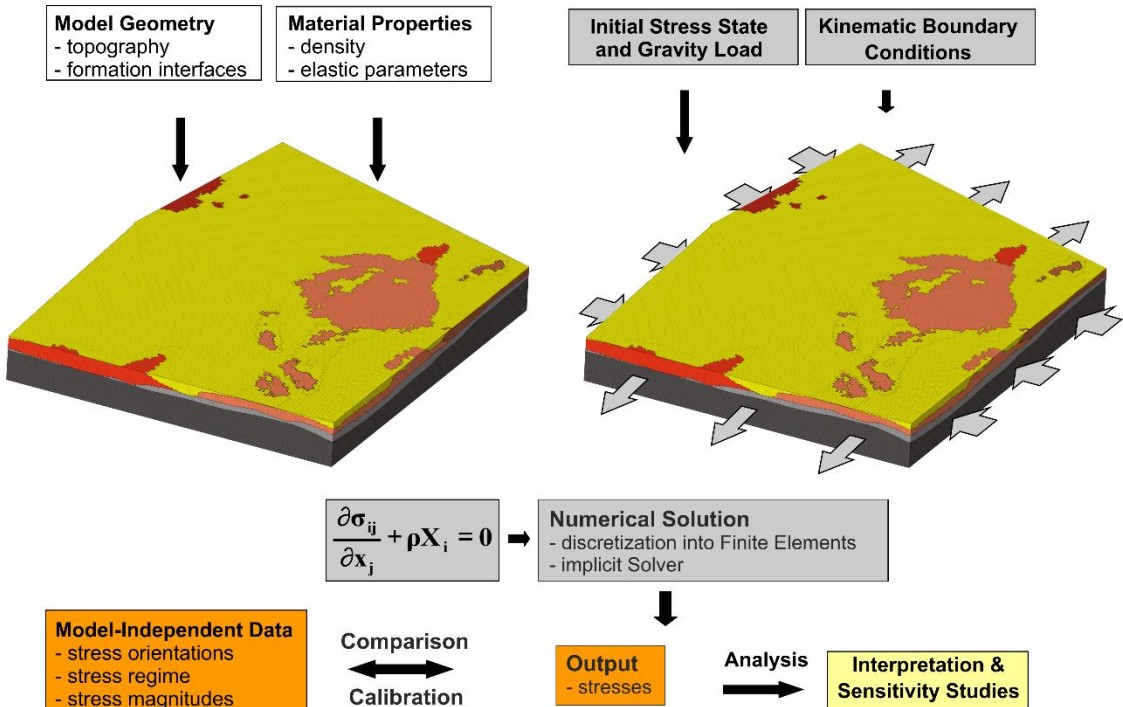

**Figure 3: General workflow of 3D geomechanical-numerical modelling. White boxes: Assembly of model geometry and rock properties. Left figure: 3D view of the discretized model volume. Grey boxes: Initial stress field and kinematic boundary**
**conditions, gravity load and numerical solution. An appropriate initial stress state and kinematic boundary conditions are determined and applied as well as gravity load. Right figure: Discretized model volume including boundary conditions used. The partial differential equation of the equilibrium of forces in 3D is solved using the FEM ($\sigma_{ij}$ stress tensor, $x_j$ Cartesian coordinates, $\rho$ density, and $X_i$ body forces). Orange boxes: Model results are calibrated against model-independent observations. Yellow box: Once the fit to the model-independent observations is acceptable, i.e. within their uncertainties, an interpretation and analysis of**
**the model results can be performed.**

### 3.2 Model geometry

The model geometry extends over 1250 km in east-west direction from eastern Poland to western France and by 1000 km in north-south direction from southern Scandinavia to northern Italy covering an area of about 1.25 million km$^2$. This area was chosen with regard to the orientation of $S_{Hmax}$ to simplify the definition of boundary conditions later on and with regard to

important crustal structures which may affect the recent stress field in Germany, e.g. the Bohemian Massif, the Avalonia-EEC suture and the European Cenozoic Rift System. Additionally, model boundaries are selected distal to the German border to avoid possible boundary effects in the area of main interest. (Fig. 1)

The model geometry contains seven units: A sedimentary cover, the upper crust subdivided into four units, the lower crust and parts of the lithospheric mantle. The units are bounded by five surfaces: The topography, the top of the crystalline basement, the top of the lower crust, the Mohorovičić discontinuity (Moho) and the model base at 100 km depth. The bottom of the model is thus not defined as the LAB and therefore the thickness of the lithospheric mantle can deviate from its real thickness. The Moho was chosen as the deepest surface since almost all calibration data are from above and also the depth interval of greatest interest are the upper 10 km of the crust. Although deeper structures may exert a long-wavelength effect on the stress state in the upper crust we expect that the primary contributions to the stress field are captured by the considered interfaces. The upper crust is laterally sub-divided into four parts: The EEC, Avalonia, the Armorican Terrane Assemblage and the ALCAPA unit referring to the tectonic units displayed in Fig. 1d.

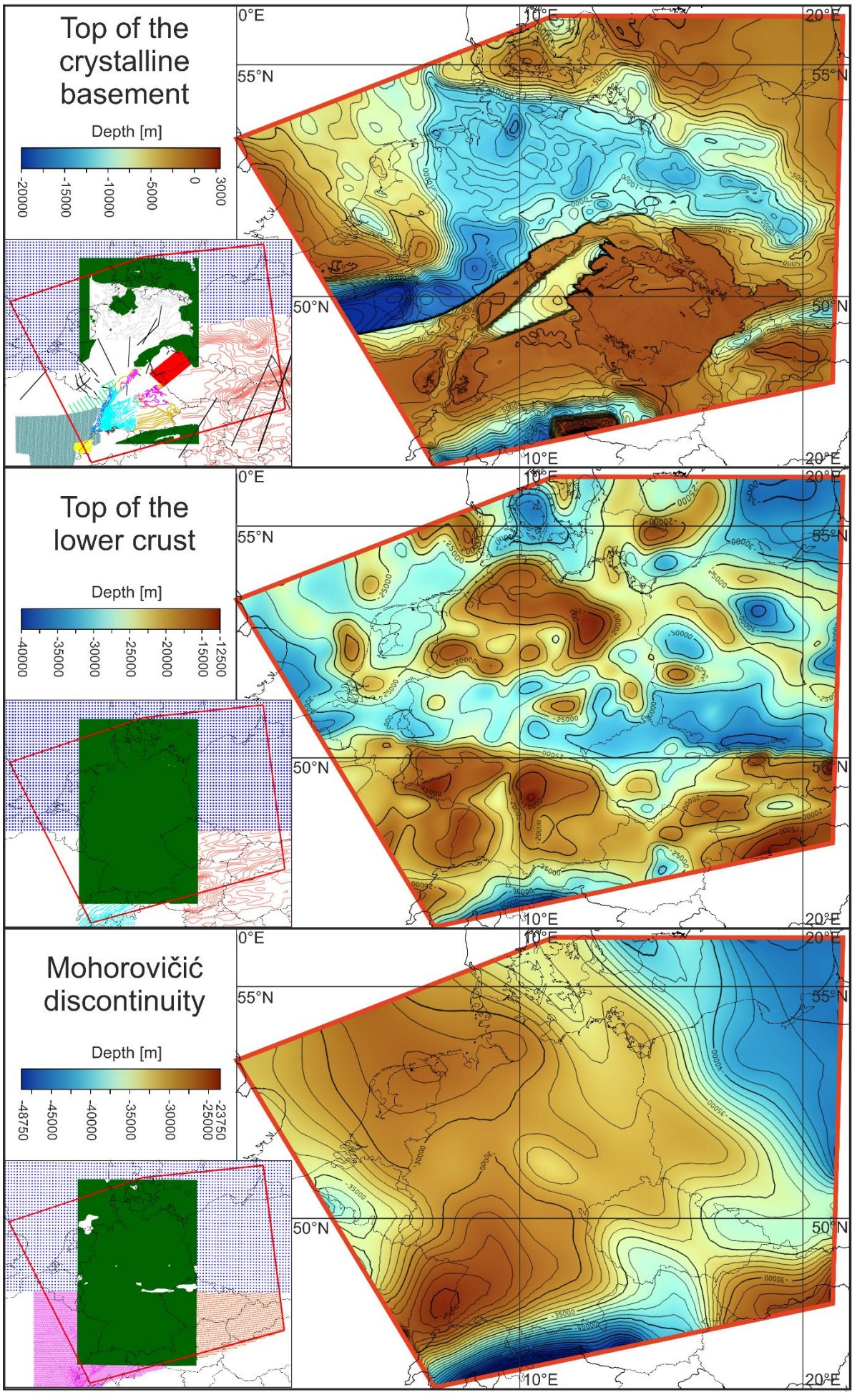

**Figure 4: Database and depth maps of the top of the crystalline basement, the top of the lower crust and the Mohorovičić discontinuity. We used the color map 'roma' of Crameri (2021). Used data are: Crystalline basement: Anikiev et al., 2019 (dark green), Diebold et al., 1991 (brown), GeORG-Projektteam, 2013 (light blue), Geothermieatlas Bayern, 2004 (pink), Hurtig et al., 1992 (blue green), Kirsch et al., 2017 (red), Korsch and Schäfer, 1995 (light green), Lindner et al., 2004 (grey), Maystrenko and Scheck-Wenderoth, 2013 (dark blue), Reinhold, 2005 (orange), Rupf and Nitsch, 2008 (cyan), Sommaruga, 1999 (yellow), Tašárová et al., 2016 (light red), black profiles (Behr et al., 1994; Bokelmann and Bianchi, 2018; Cazes et al., 1985; Freeman and Mueller, 1992; Grad et al., 2009a; Heinrichs et al., 1994; Hirschmann, 1996; Janik et al., 2011; Meschede and Warr, 2019; Oncken et al., 2000; Reinhold, 2005; Schintgen, 2015; Wenzel and Brun, 1991). A labeled, larger size map of this database is available in the supplementary data. Top of the lower crust: Anikiev et al., 2019 (dark green), Maystrenko and Scheck-Wenderoth, 2013 (dark blue), Tašárová et al., 2016 (light red), Valasek and Mueller, 1997 (cyan). Mohorovičić discontinuity: Anikiev et al., 2019 (dark green), Grad et al., 2009b (red); Maystrenko and Scheck-Wenderoth, 2013 (dark blue); Wagner et al., 2012 (pink).**

Figure 4 shows the depth maps of the top crystalline basement, the top of the lower crust and the Moho with the corresponding database used. The model is mainly based on three existing models. The 3D Deutschland model (Anikiev et al., 2019) the Central European Basin model (Maystrenko and Scheck-Wenderoth, 2013) and the Central Europe model of Tašárová et al. (2016).

The key challenge was the construction of the top crystalline basement. In all the models used and also in most other datasets the base of the sedimentary layer is defined as the top of the basement regardless of whether the basement consists of crystalline or low-grade metamorphic rocks. This is an assumption which is not sufficient to represent the stiffness contrast correctly. The main reason for this assumption is the lack of data due to the usually great depths and the lack of economic interest in these units. Especially in the Rhenohercynian and Saxothuringian Zone (Fig. 1c) only a few seismic profiles exist from research projects like DEKORP (Meissner and Bortfeld, 1990), EGT (Freeman and Mueller, 1992) or ZENTROSEIS (Bormann et al., 1986). Despite the uncertainties due to this poor amount of data, the use of the sediment-crystalline boundary is necessary for a geomechanical-numerical model, because of the strongly different mechanical properties. An extreme example within our model area is the western part of the Rhenohercynian Zone. Here the basement is outcropping, e. g. in the Rhenish Massif, but the top of the crystalline crust is suspected to be at about 20 km depth (Schintgen, 2015; Oncken et al., 2000). Therefore, in those areas where the definition of the basement does not correlate to the top of the crystalline basement, we constructed this surface to obtain a mechanically uniform surface; data used are shown in Fig. 4. The boundaries between the Variscan basement units are simplified as vertical due to the poor knowledge.

## 3.3 Model discretization

Our final mesh shown in Fig. 5 comprises 1.32 million hexahedral elements with a lateral homogenous resolution of approximately $6 \times 6$ km$^2$. The vertical resolution decreases with depths from 800 m near the surface up to 7500 m at the base of the model. An exception is the uppermost element layer, which is only 50 m thin to reduce the impact of free surface effects in the uppermost units. Due to the complex geometry of our model we decided not to use the common approach in the upper units in which each unit is meshed individually. Only the mantle and the crust are meshed as whole. Then we use the tool ApplePy (Ziegler et al., 2019) to assign each finite element to the respective subunits and the appropriate rock properties.

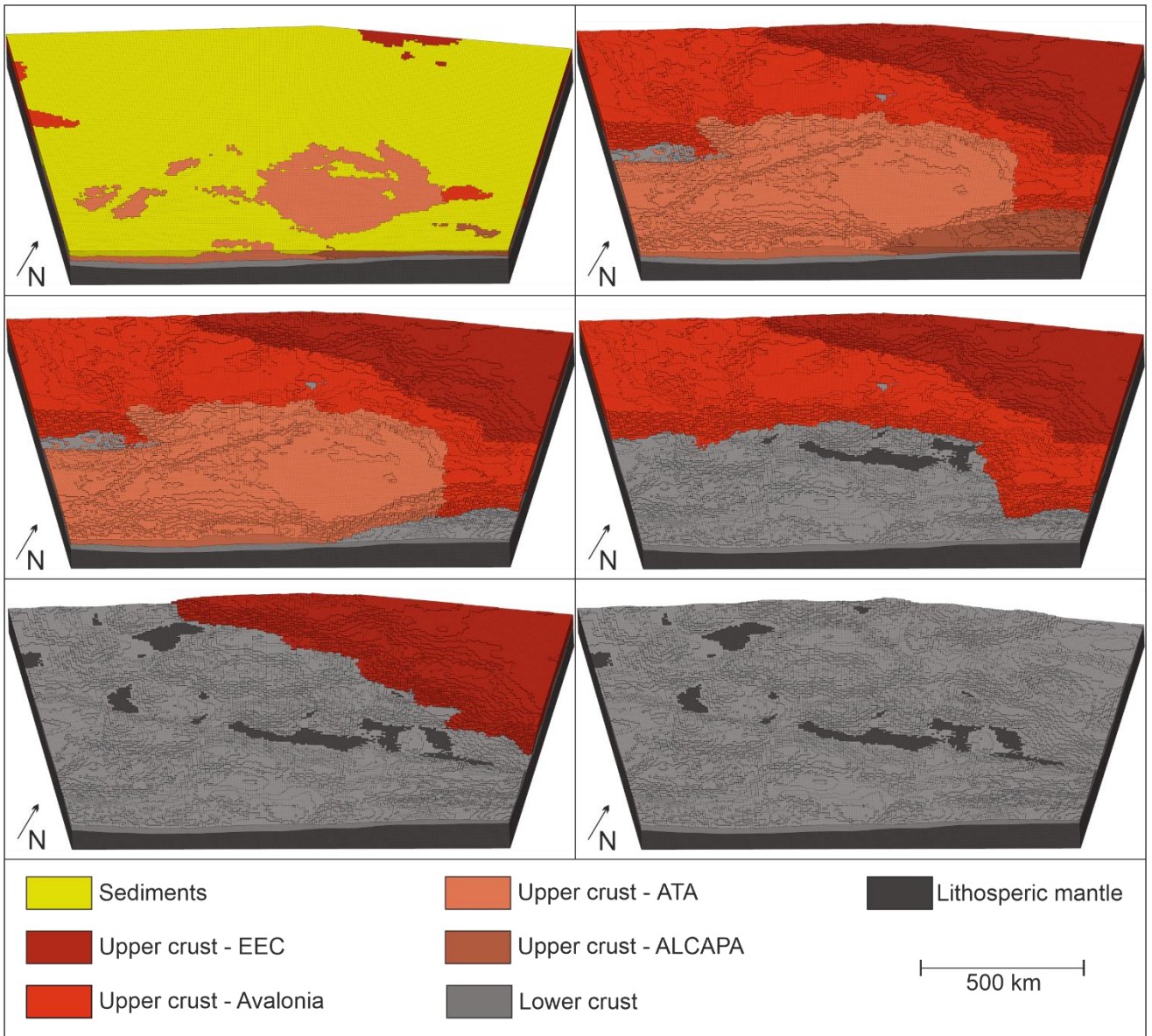

**Figure 5: Six different views of the discretized model showing the internal model structure. The sedimentary unit is colored in yellow, the upper crust in different red shades regarding to different tectonic units (Fig. 1c), the lower crust in light grey and the lithospheric mantle in dark grey. The dimension of the model is 1000 x 1250 x 100 km³ comprising 1.32 million hexahedral elements. ATA - Amorican Terrane Assemblage, EEC - East European Craton.**

## 3.4 Rock properties

The material properties used in the model and corresponding references are shown in Table 2. The assignment of mean rock properties to the sediment unit is a difficult task, due to the large number of different rock types represented by unconsolidated rocks, claystones, sandstones, salt or limestones. Therefore, the values are approximate mean values. For the upper crust we applied a different density for each tectonic unit in the range of 2750 to 2820 kg m$^{-3}$. The Young's modulus and Poisson's ratio are values for granodiorite as the characteristic rock of the upper crust. For the lower crust density we use the results of Maystrenko and Scheck-Wenderoth (2013) and Tašárová et al. (2016) but since, unlike them, we have only one uniform unit, we use an average of 3000 kg m$^{-3}$. The Young's modulus and the Poisson's ratio are again values for the characteristic rock of the unit, in this case gabbro.

**Table 2: Overview of the parameters used for the parametrization. [a]Turcotte and Schubert (2014), [b]Maystrenko and Scheck-Wenderoth (2013), [c]Tašárová et al. (2016), [d]Przybycin et al. (2015)**

| Unit | Density [kg m$^{-3}$] | Young's modulus [GPa] | Poisson's ratio [-] |
|---|---|---|---|
| Sediments | 2300 | 30 | 0,25 |
| Upper crust | | | |
| ALCAPA | 2750[b] | 70[a] | 0,25[a] |
| Amorican Terrane Assemblage | 2790[c] | 70[a] | 0,25[a] |
| Avalonia | 2820[b] | 70[a] | 0,25[a] |
| East European Craton | 2810[b] | 70[a] | 0,25[a] |
| Lower crust | 3000[b,c] | 80[a] | 0,25[a] |
| Lithospheric mantle | 3300[c,d] | 130 | 0,28[a] |

### 3.5 Initial stress state

Before applying displacement boundary conditions to the model an initial stress state is generated representing a reference stress state. We use a simple semi-empirical function by Sheorey (1994) for the stress ratio k depending on depth (z) and the Young's modulus (E) which can be considered as being representative for tectonically inactive regions with low lateral

density contrasts:

$$k = 0.25 + 7E \left( 0.001 + \frac{1}{z} \right) \qquad (1)$$

To achieve our initial stress state we compare the k values defined as

$$k = \frac{S_{Hmean}}{S_V} = \frac{S_{Hmax} + S_{hmin}}{2\, S_V} \qquad (2)$$

from 29 synthetic profiles with the stress ratio calculated for a Young's modulus of 30 and 70 GPa, representing the

260 sedimentary and upper crust units.

In order to establish the initial stress state, an underburden and a sideburden are added and this extended model is implemented in a conic shell (Fig. 6a and b). Then, the model has to settle down frictionless within this conical shell. During that procedure, the Young's modulus in the underburden as well as the Poison's ratio in all units is varied until the virtual wells fit the Sheorey equation (Eq. 1, Fig. 6c). This procedure has been used and described several times (Buchmann and

265 Connolly, 2007; Hergert, 2009; Hergert and Heidbach, 2011; Reiter and Heidbach, 2014). The resulting stress state represents the initial stress state, which is subsequently perturbed by applying displacement boundary conditions that impose the tectonic stress.

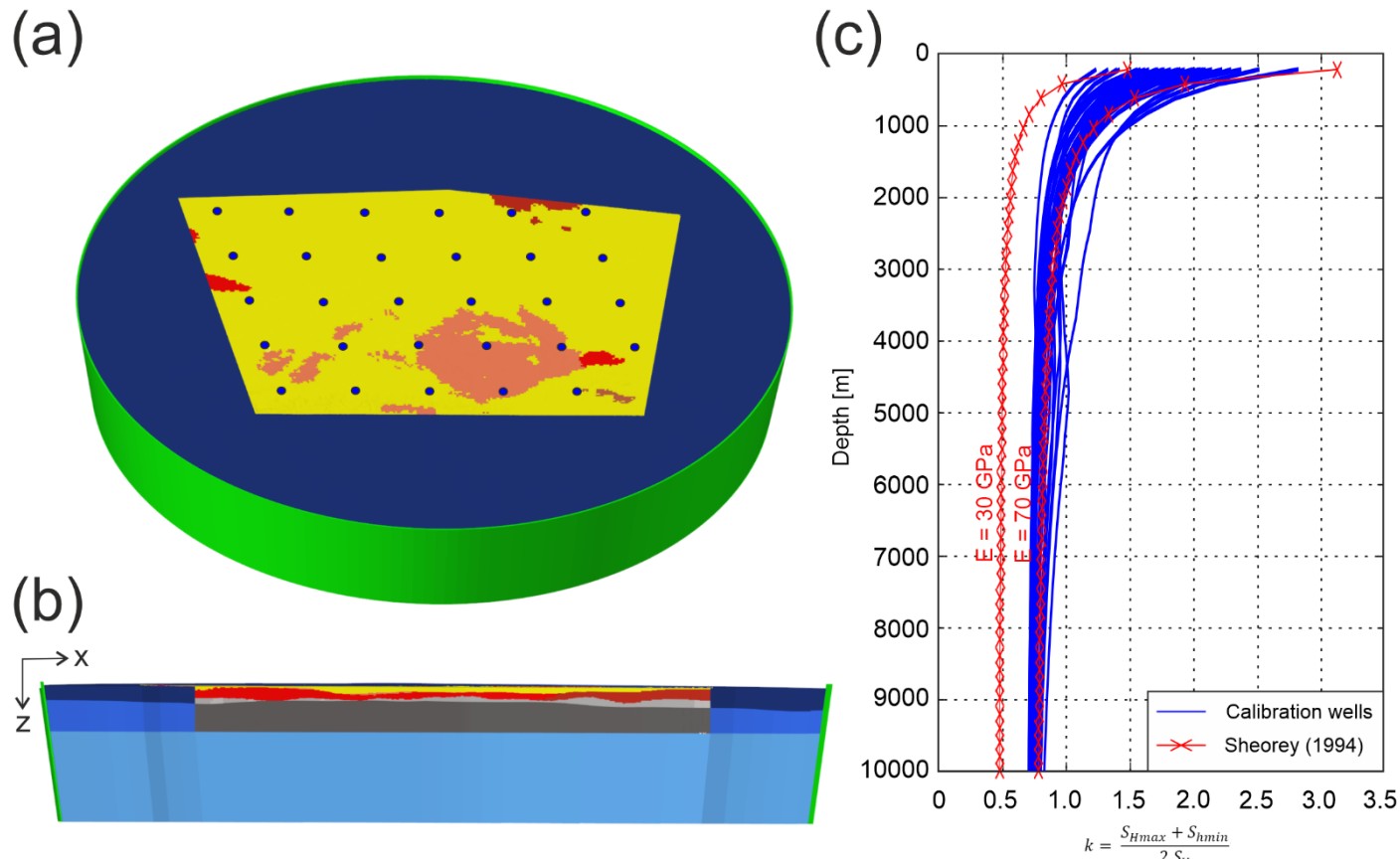

**Figure 6: (a) Top View of the model implemented in the shell (green) and the sideburden (dark blue). Blue dots indicate the synthetic calibration wells. (b) Side view of the model implemented in the shell, the sideburden and the underburden (bright blue) (c) k-values of the calibration wells (blue curves) in comparison with k-values calculated with a semi-empirical function by Sheorey (1994) for a Young's modulus of 30 and 70 GPa representing the sedimentary and the upper crust units (red curves).**

### 3.6 Displacement boundary conditions

The base of the model is fixed vertically, lateral movements are allowed and the model surface is free. At the five lateral boundaries of the model displacement boundary conditions are applied to parametrize past and ongoing tectonic kinematics. The orientations of the model boundaries are chosen parallel or perpendicular to the mean $S_{Hmax}$ orientation (Fig 1b). The eastern and western lateral model boundaries are aligned parallel and the northern and southern boundaries perpendicular to the mean $S_{Hmax}$ orientation. Accordingly, extension is applied to the eastern and western boundaries and shortening to the northern and southern ones (Fig. 3).

We use a two-stage approach to find a good agreement with the stress orientation and stress magnitude datasets. First a best-fit with respect to a mean $S_{Hmax}$ orientation (see details in Sect. 4.1) is estimated by an appropriate ratio between the extension and shortening applied. In a second step we vary the magnitude of these displacements on the model boundaries while keeping the ratio constant, so that a best-fit with the stress magnitude data is achieved as well. The calibration is mainly based on the $S_{hmin}$ magnitude due to the larger amount of data from the compilation of Morawietz et al. (2020) and the fact that $S_{Hmax}$ magnitudes are often calculated and not measured and therefore less reliable. For the best-fit model a total extension of 465 m in east-west direction and a total shortening of 325 m in north-south direction is applied.

## 4. Results

### 4.1 Orientation of the maximum horizontal stress ($S_{Hmax}$)

We compare our model results with the stress orientation from the WSM database (Heidbach et al., 2018) and some additional data by Levi et al. (2019) from western Austria (Fig. 1a). From the WSM database we use only $S_{Hmax}$ orientations

that have a WSM quality A to C. However, we do not use individual data records, but a mean $S_{Hmax}$ orientation on a regular 0.5° grid (Fig. 1b and 7c). Using a mean $S_{Hmax}$ orientation avoids effects of data clustering which is often the case in the WSM database and it filters the data for a wavelength of the stress pattern that is representative for the resolution of the model. For the estimation of the mean $S_{Hmax}$ values we use the tool stress2grid from Ziegler and Heidbach (2017). The $S_{Hmax}$ data records are weighted according to their quality and their distance to the grid points. Each grid point requires at least ten data points within a fixed search radius of 200 km. The resulting mean orientation of $S_{Hmax}$ has a median standard deviation of ~28° using the statistics of bi-polar data (Mardia, 1972). The model results are interpolated linearly on a plane at 5 km depth and then the nearest value to each grid point is chosen for the comparison with the mean WSM data.

Figure 7a displays the $S_{Hmax}$ orientation of the model at 5 km depth, whereas Fig. 7c shows the calculated mean $S_{Hmax}$ orientation of the WSM data within the model area. The modelled $S_{Hmax}$ orientations at the model boundaries are controlled by the assigned boundary conditions, thus the orientations are perpendicular to the northern and southern boundaries and parallel to the eastern and western edges. Within the model area the orientation of $S_{Hmax}$ shows a homogenous pattern with a dominant NNW-SSE orientation which rotates slightly to a north-south orientation at the eastern boundary. Figure 7b visualizes the deviation of the model results from the mean WSM data. Blue indicates regions where the model results are rotated anti-clockwise with respect to the mean WSM data and orange regions with a clockwise rotation. There are three areas with larger deviations. One with primarily clockwise rotation in the area of Belgium. The two other areas, located in the northern and south-western part of the model, including the NGB, the eastern part of the Alps and western part of the Carpathians show an anti-clockwise rotation. Apart from these two areas, the dominant color is orange, conterminous with a slight clockwise rotation. This trend is also visible in Fig. 7d where the histogram of the deviation between the mean $S_{Hmax}$ orientation derived from the WSM data and the modelled orientation is shown with a median deviation of 5.6° and a mean deviation, calculated from the absolute differences, of 15.6°.

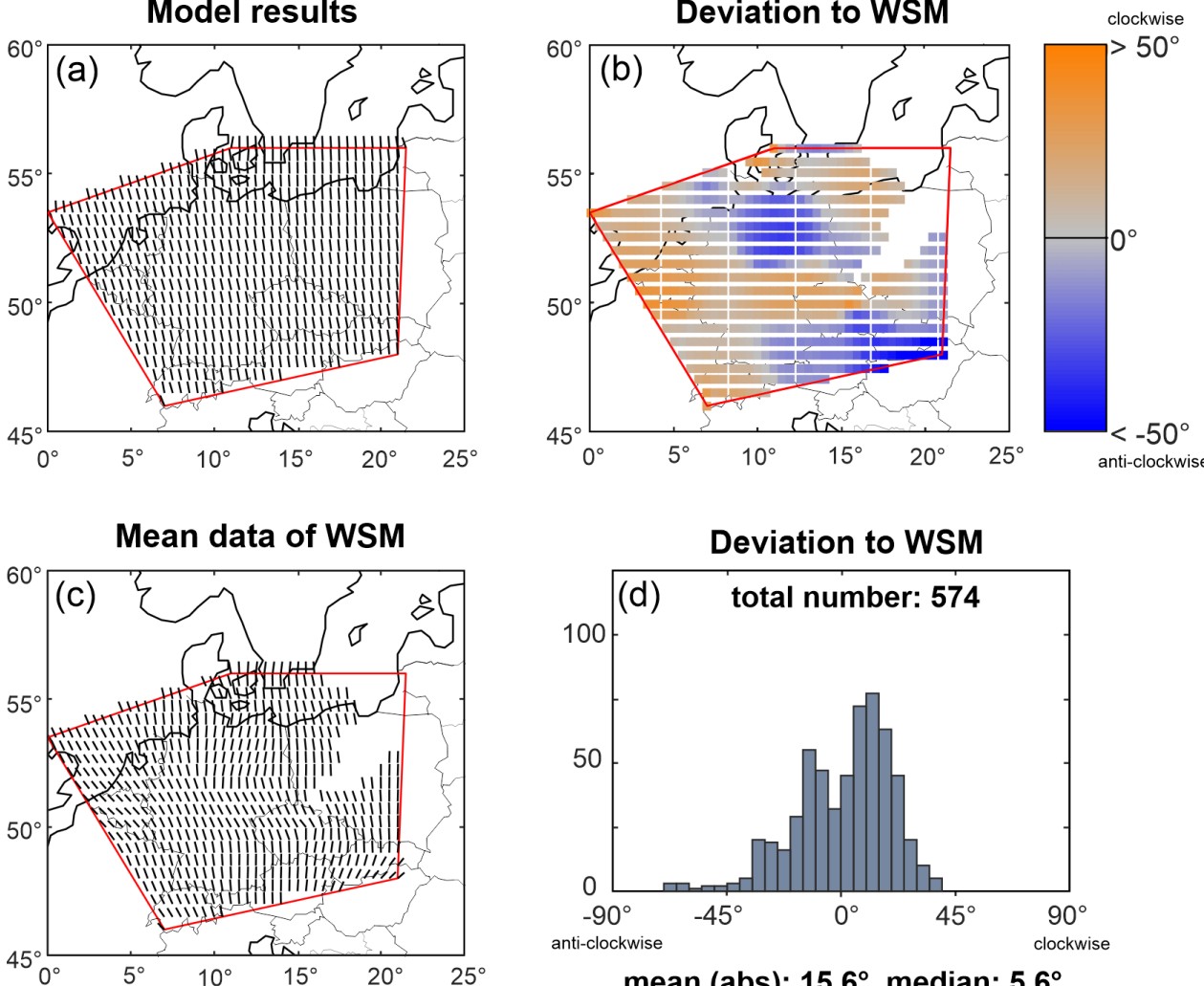

**Figure 7: Comparison of $S_{Hmax}$ orientation of the model results with the mean $S_{Hmax}$ orientation derived from WSM data. (a) $S_{Hmax}$ orientation of the model at 5 km depth. (b) Deviation of the model result relative to the mean $S_{Hmax}$ orientation derived from WSM data. (c) Orientation of the mean $S_{Hmax}$ of WSM data (details are described in the text). (d) Histogram of the deviation of the modelled $S_{Hmax}$ orientation to the mean $S_{Hmax}$ orientation derived from WSM data.**

## 4.2 Stress magnitudes

### 4.2.1 Minimum horizontal stress ($S_{hmin}$) magnitudes

The modelled magnitudes of $S_{hmin}$ in comparison to stress magnitude data of Morawietz et al. (2020) are shown in Fig. 8. The figure is divided into three subfigures displaying the differences depending on depth and quality (Fig. 8a), the spatial distribution of the calibration data (Fig. 8b) and a histogram showing the distribution of the differences between the modelled and observed $S_{hmin}$ magnitudes (Fig. 8c). The differences are calculated as interpolated model results minus data, thus positive differences correspond to too large model values and negative ones to too low model values. We use only data from Morawietz et al. (2020) with a quality of A, B and C and from depths >200 m, to avoid topographic effects. Thus, we use 74 $S_{hmin}$ magnitude data records from a depth of 200 to 4600 m, most of them from the upper 1000 m. As shown in Fig. 8b the data are mainly located within the south-western part with the exception of one measurement from the NGB. With 42 data records, more than half of all data records originate from three localities: From Falkenberg near the German-Czech border (Baumgärtner et al., 1987), from Benken in Switzerland (nagra, 2001) and from Wittelsheim in eastern France (Cornet and Burlet, 1992). Due to the calibration process described in Sect. 3.6 a median difference of 0 MPa is achieved. The differences are, with two exceptions, in a range of -10 to 10 MPa and seem to be independent of depth and quality. This together with a mean difference of 3.3 MPa indicates a very good fit with the data of Morawietz et al. (2020).

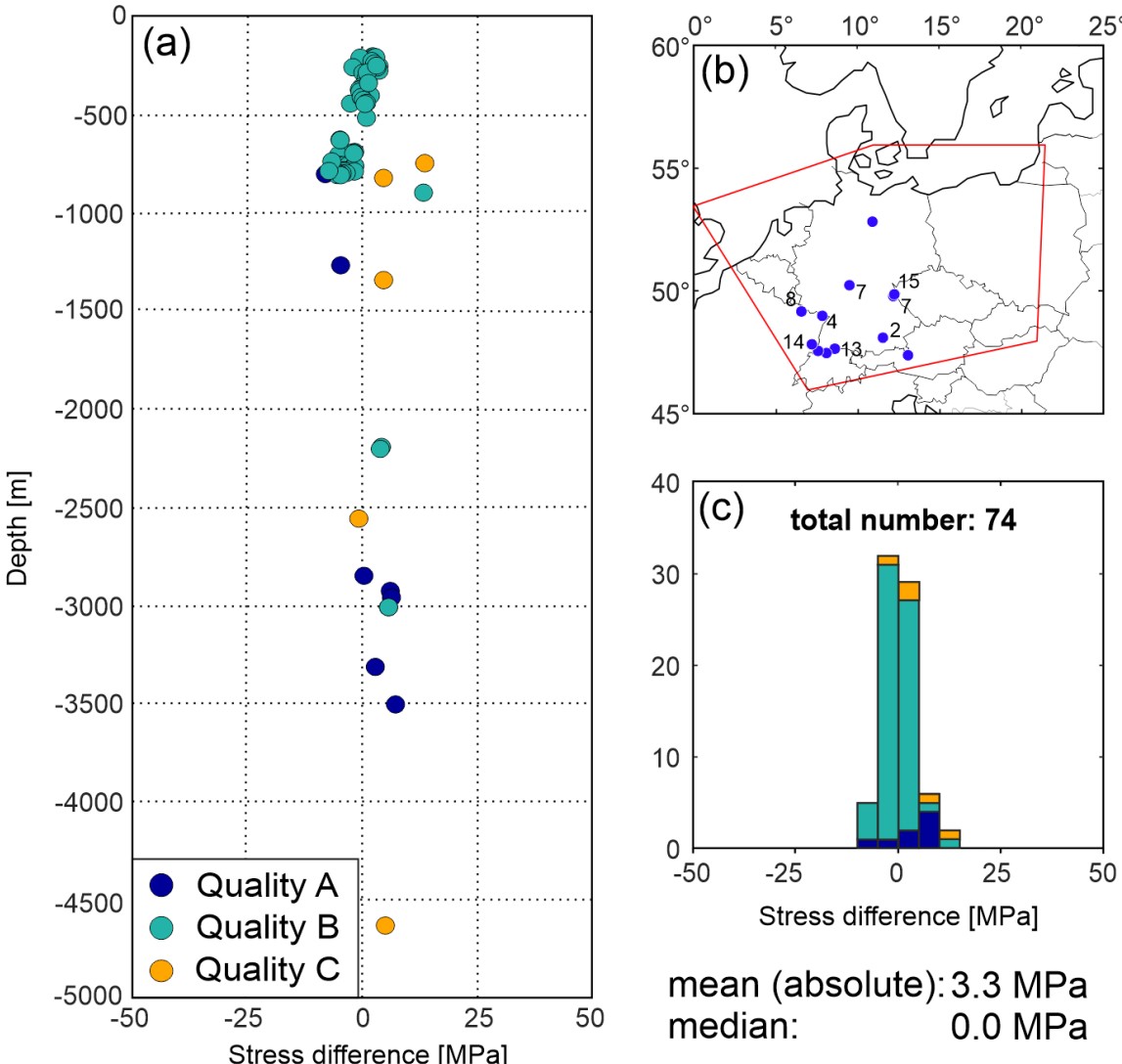

**Figure 8: S<sub>hmin</sub>** magnitudes of the model in comparison to the data of Morawietz et al. (2020). The differences are calculated as model results minus calibration data. (a) Depth depending differences. Color of dots indicates the quality of the calibration data. (b) Spatial distribution of the calibration data used, numbers indicating localities with multiple data used. (c) Histogram of the differences displayed in (a).

### 4.2.2 Maximum horizontal stress ($S_{Hmax}$) magnitudes

For the model calibration regarding the $S_{Hmax}$ magnitudes 57 data records are used from the database of Morawietz et al. (2020). Again, only data with a quality of A to C and from a depth of >200 m are used. Similar to the $S_{hmin}$ data they are mainly located in the south-western part of the model area (Fig. 9b). The data are from seven different localities, whereby the data from Falkenberg near the German-Czech border (Baumgärtner et al., 1987) and from Benken in Switzerland (nagra, 2001) with 25 data records make up almost half of the comparison data used. The mean of the absolute difference is 20.6 MPa and the median difference is 19.3 MPa. This difference can be explained by the asymmetric depth distribution of the values (Fig. 9a). There are significantly more data records from shallower depths (200 to 1000 m) which indicate too large model results than from the greater depths (>1000 m) which indicate too low results. Regardless of this, a trend is visible from positive to negative stress differences with increasing depth, i.e. the model seems to predict too large values of $S_{Hmax}$ in the upper part of the model and too low values of $S_{Hmax}$ in the deeper part. Furthermore, it is striking that the differences of quality A data are almost all negative and almost all of quality B and C are positive.

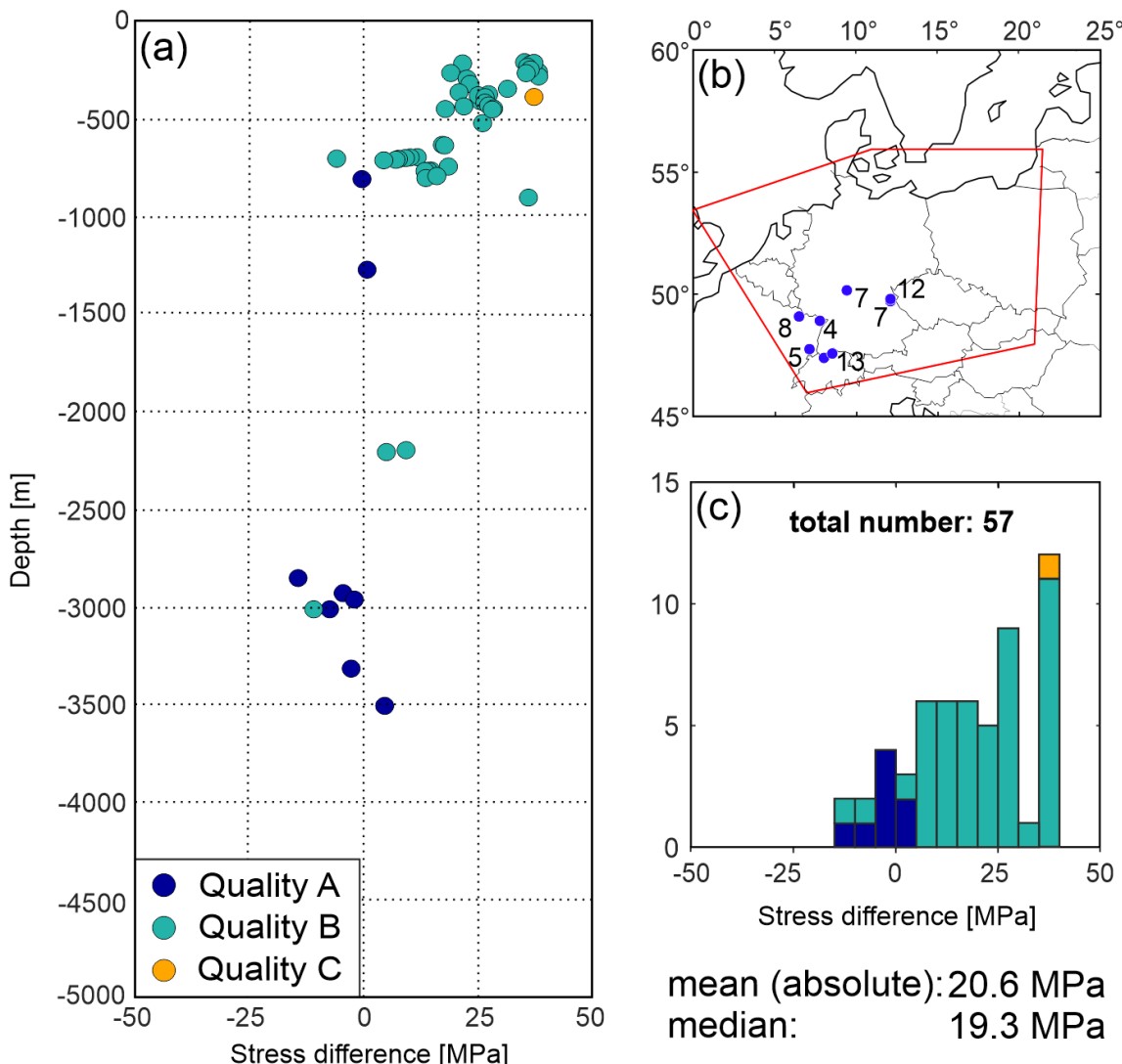

**Figure 9:** **S_Hmax magnitudes of the model in comparison to the data of Morawietz et al. (2020). The differences are calculated as model results minus calibration data. (a) Depth depending differences. Color of dots indicates the quality of the calibration data. (b) Spatial distribution of the calibration data used, numbers indicating localities with multiple data used. (c) Histogram of the differences displayed in (a).**

### 4.2.3 Stress gradients and stress regime

Additional to the calibration of the model with stress magnitude data, the absolute stress magnitudes of $S_V$, $S_{hmin}$ and $S_{Hmax}$ for three hypothetical wells up to 10 km depth are shown in Fig. 10. We have chosen these three locations (Fig. 11) due to the availability of stress data for a comparison later on, the quite uniform distribution over Germany (north, south-west and

360 south-east) and the different depths of the crystalline surface. The hypothetic well 1 is entirely within the crystalline basement, well 2 entirely within the sedimentary unit and well 3 partly within the sediment unit and partly within the crystalline basement. As with the previous results we do not show the results of the upper 200 m. The depths are relative to the model surface and do not correspond to the z values of the model.

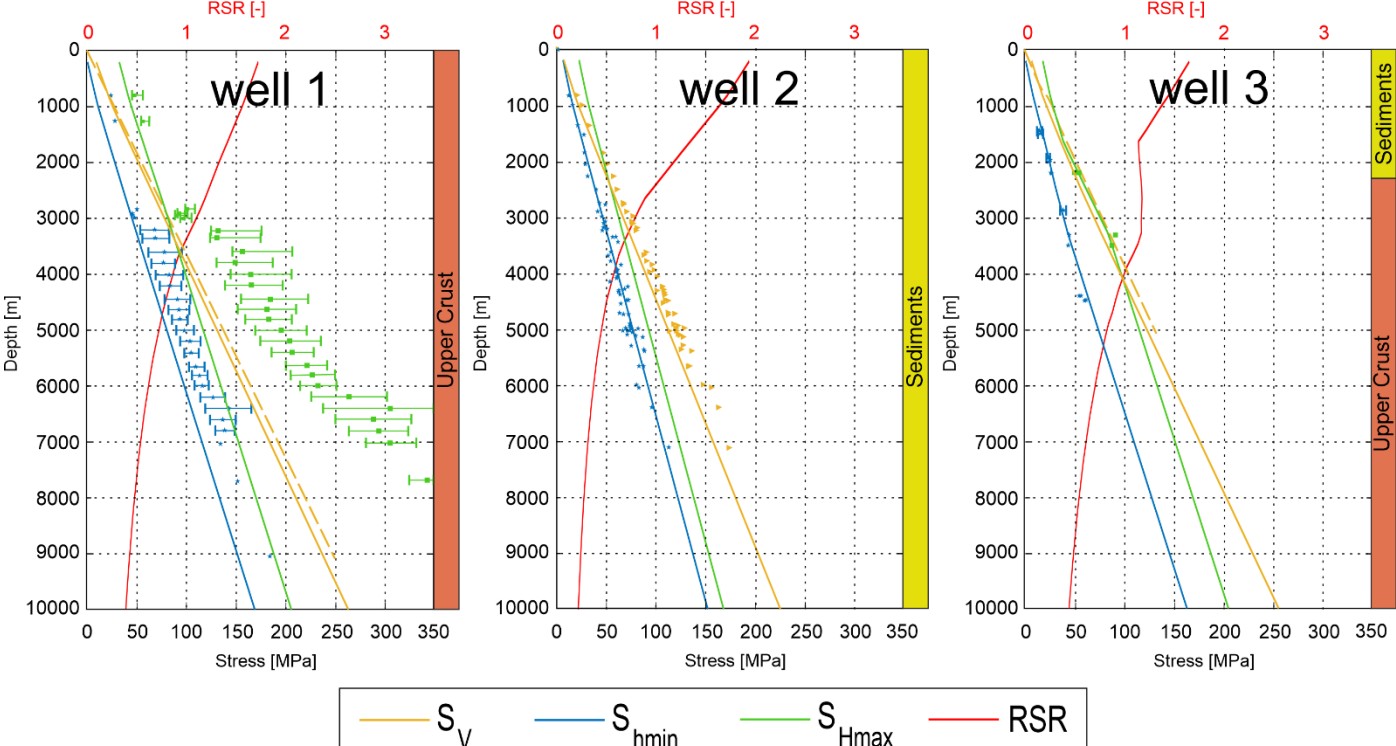

Figure 10: Gradients of regime stress ratio (RSR), $S_V$, $S_{hmin}$ and $S_{Hmax}$ for three hypothetic wells in comparison with data. The orientation of $S_{Hmax}$ (not shown here) is constant over the entire depth (well 1: 161°, well 2: 163°, well 3: 162°). The colored lines show the RSR (red), the stresses of $S_V$ (yellow), $S_{hmin}$ (blue) and $S_{Hmax}$ (green). Blue stars and green rectangles show measured respectively calculated magnitude of $S_{hmin}$ and $S_{Hmax}$. The uncertainties of the magnitudes if specified are displayed as error bars. Yellow dotted lines are calculated $S_V$ from density data. Well 1: Comparison data from the 'Kontinentale Tiefbohrung' (KTB) of Brudy et al. (1997). Well 2: Comparison data from the NGB of Röckel and Lempp (2003). Well 3: Comparison data from Soultz-sous-Forêts and Rittershofen. Measured $S_{hmin}$ magnitudes of Valley and Evans (2007), calculated $S_{Hmax}$ values of Klee and Rummel (1993) and calculated $S_V$ magnitudes based on density values of Azzola et al. (2019).

The $S_V$ gradients of well 1 and 2 are constant with the exception of a slight increase in the upper 1000 m, which is related to free surface effects. Well 3 also shows this effect but with an additional gradient change at 1500 to 3500 m depth. Above 1500 m depth the gradient corresponds to well 2 and below 3500 m to well 1. Overall well 2 shows the lowest $S_V$ gradient of about 22.5 MPa km$^{-1}$, well 1 the highest $S_V$ gradient of about 27 MPa km$^{-1}$ and well 3 is with ~25.5 MPa km$^{-1}$ in between. The horizontal stresses of $S_{hmin}$ and $S_{Hmax}$ have almost constant gradients in well 1 and 2, only the absolute stresses differ. In well 1 we have a gradient of about 17 MPa km$^{-1}$ resulting in 170 MPa at 10km depth for $S_{hmin}$ and about 205 MPa at 10 km for $S_{Hmax}$. Due to the identical gradients the differential stress between $S_{Hmax}$ and $S_{hmin}$ is constant 35 MPa all over the well path. Well 2 show a similar pattern with a $S_{hmin}$ and $S_{Hmax}$ gradient of about 15 MPa km$^{-1}$ and a constant differential stress between $S_{hmin}$ and $S_{Hmax}$ of about 15 MPa. The gradients of well 3 are, as with the $S_V$ magnitudes, a combination of well 1 and 2. This can be seen particularly clearly by the $S_{hmin}$ values of well 3. From the surface to a depth of ~1500 m the gradient is quite similar to well 2 and below ~3500 m to well 1. In between the gradient increases to ~25 MPa km$^{-1}$ which is the highest gradient of all horizontal stresses displayed. As a result, the differential stress in well 3 between $S_{Hmax}$ and $S_{hmin}$ also changes with depth. It amounts to 20 MPa at 1500 m depth, increasing with depth to about 40 MPa at 4000 m and then remains constant leading to 165 MPa for $S_{hmin}$ and 205 MPa for $S_{Hmax}$ at 10 km depth. Well 3 shows thus the only significant change of horizontal differential stress with depth of all three wells shown and also the highest differential stress with a maximum of about 40 MPa below 4000 m depth.

All three wells show a change of the stress regime from strike-slip to normal faulting, with $S_V$ becoming greater than $S_{Hmax}$. In well 1 the transition is at about 3500 m, in well 2 at about 2500 m and in well 3 at about 4000 m depth. But despite these minor differences in depth, there are almost no differences between the stress regimes.

As an additional result the regime stress ratio (RSR) (Simpson, 1997) for four model sections and for the three wells are shown in Fig. 10 and 11. The RSR (Eq. 3) is a unitless value combining the stress regime index n (Eq. 4) of Anderson (1905) and the ratio of stress differences $\phi$ (Eq. 5):

$\quad RSR = (n + 0.5) + (-1)^n(\phi - 0.5)$ (Simpson, 1997) $\hfill$ (3)

$$n = \begin{cases} 0 & S_{hmin} < S_{Hmax} < S_V \\ 1 & S_{hmin} < S_V < S_{Hmax} \\ 2 & S_V < S_{hmin} < S_{Hmax} \end{cases} \text{ (Anderson, 1905)} \hfill (4)$$

$$\phi = \frac{(\sigma_2 - \sigma_3)}{(\sigma_1 - \sigma_3)} \text{ (Angelier, 1979)} \hfill (5)$$

The resulting value between 0 and 3 is the RSR indicating the stress regime divided into 6 classes from radial extension over extension, transtension, transpression and compression to constriction. The results in Fig. 11 show with the exception of
400 peripheral areas, displaying some boundary effects, a rather uniform change of the stress regime from strike-slip to normal faulting with increasing depth. Starting with a RSR of 1 to 2 at 1000 m resulting in a RSR of 1 to 0.25 at 10 km depth. The two sections in between at 2000 and 4000 m show the transition with a dominant RSR of 1 to 1.5 and 0.75 to 1.25, respectively. The RSR of the three wells displayed in Fig. 10 confirm this observation. In the shallower parts the RSR lies between 1.75 and 2 then decreasing with depth to values smaller than 0.5. A special aspect is visible in well 3, where the
405 RSR is almost constant over 2 km along the transition between the sedimentary and the upper crust unit. The lowest RSR occurs in well 2 which is located entirely within the sedimentary unit. This correlation is also visible in Fig. 11 where the lower RSR is related to area with high sediment thicknesses, e.g. the NGB. On the other hand, a higher RSR seems to correlate with basement areas like the Bohemian Massif or the Mid German Crystalline High, well visible at 2000 and 4000 m depth.

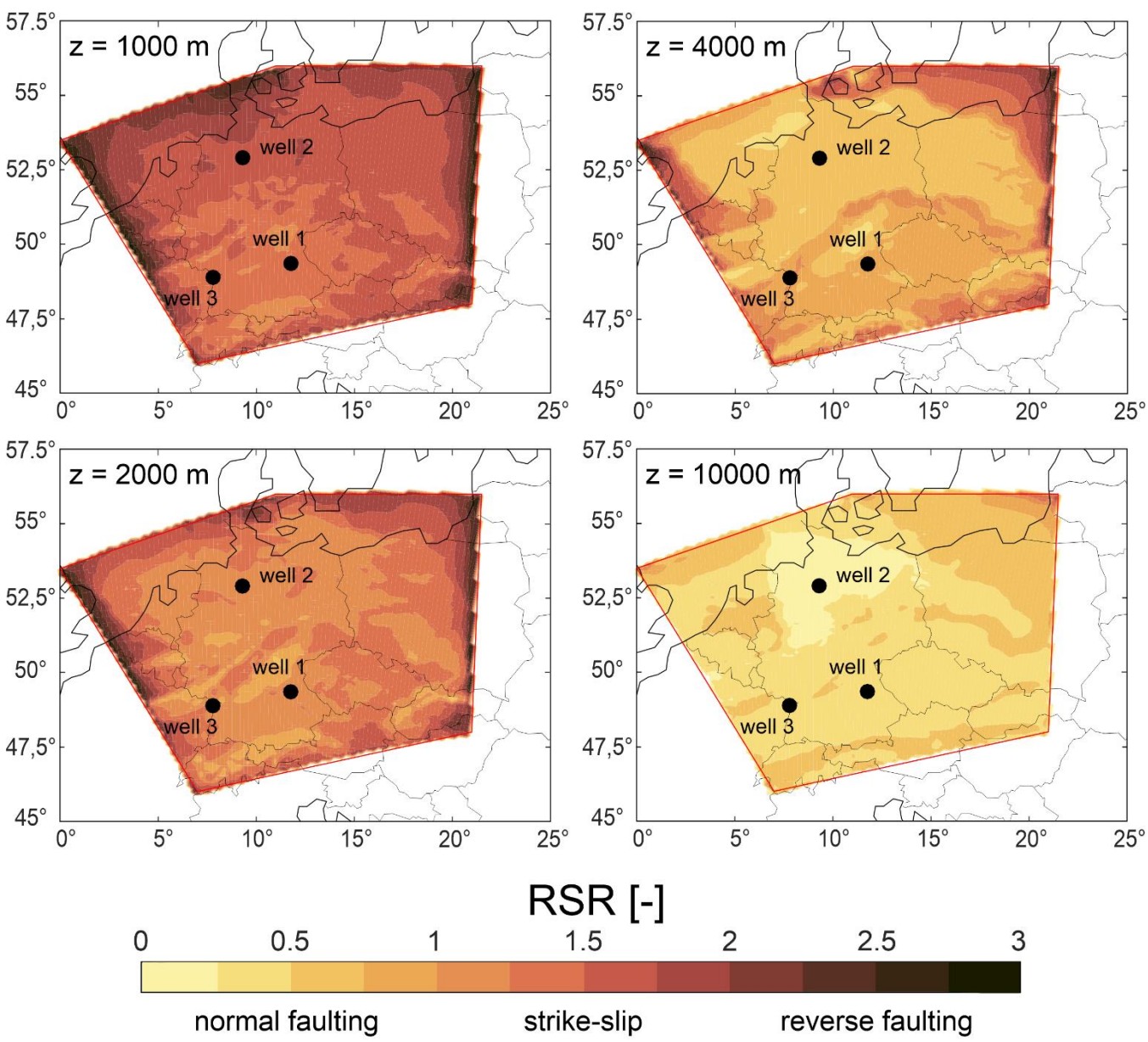

**Figure 11: RSR indicating the stress regime for four model sections for different depths. The three black dots show the locations of the three hypothetical wells in Fig. 10. The high RSR values at the model edges in the upper 4000 m, representing a constriction, are edge effects due to the applied boundary conditions. We used a color map based on 'lajolla' of Crameri (2021).**

## 5. Discussion

### 5.1 Orientation of the maximum horizontal stress ($S_{Hmax}$)

The results of the $S_{Hmax}$ orientation are in comparison to the mean WSM data quite good with a median deviation of 5.6° and an absolute mean deviation of 15.6° (Fig. 7). Therefore, the results are within the error range of the used WSM A to C quality data records which have uncertainties of 15 to 25° (Heidbach et al., 2018). Apart from some reorientation at the model edges in the upper 100s of meters, the orientations are almost constant over the entire model depth. For example, the $S_{Hmax}$ orientation of our three hypothetical wells are 161°, 163° and 162° (Fig. 10).

However, there are three areas with noticeable lateral deviations: In the north-eastern part of Germany, in Belgium and along the Carpathians. The region in north-eastern Germany belongs to the NGB in which there are thick salt deposits. Salt can act as a mechanical decoupling horizon between the layers above and below (Ahlers et al., 2019; Bell, 1996; Cornet and Röckel, 2012; Heidbach et al., 2007; Hillis and Nelson, 2005; Tingay et al., 2011). In such cases, the stress state below represents the regional trend transferred through the crust while the stress state above is only affected by local sources often controlled by

local density and strength contrasts. More than 20 % of the data from this region are above the salt and mostly E-W oriented, in contrast to the data below, which are more N-S oriented (e.g. Cornet and Röckel, 2012; Grote, 1998; Röckel and Lempp, 2003; Roth and Fleckenstein, 2001). However, since we do not distinguish between the data from these different layers the derived mean $S_{Hmax}$ values are influenced by these data above the salt layer. Possibly the misfit in this area can also be explained by the Pritzwalk anomaly, a positive gravity anomaly due to high-density lower crust (Krawczyk et al., 2008).


The deviations in Belgium and adjacent areas can have several reasons. This region is the border between two massifs, the Rhenish and the Brabant Massif (Pharaoh, 2018), and the strength contrast between these two massifs may play a role. Such contrasts are often considered to be responsible for reorientations in the stress field (e.g. Adams and Bell, 1991; Heidbach et al., 2007; Rajabi et al., 2017). Another reason could be the tectonically active Lower Rhine Basin nearby or the uplift of the Rhenish Massif (Reicherter et al., 2008). Possible boundary effects can be excluded, since the orientation of $S_{Hmax}$ is uniform


along the entire western boundary of the model. The deviations in the south-eastern part of the model are located along the Carpathians and the adjacent Pannonian Basin. This is possibly an area with low far-field or first-order horizontal stress sources resulting in a near isotropic stress state (Heidbach et al., 2007) and thus the topography contrast between the mountain range and the Pannonian Basin has probably a dominant influence (Bada et al., 2001; 1998). Furthermore, the NE-


SW oriented $S_{Hmax}$ indicated by the WSM data implies an NW-SE extension in this area which is in agreement with the orientation of back-arc extension arising from the retreating slab beneath the Carpathians in Romania (Sperner et al., 2001), which the model does not account for.

In general, our modelled $S_{Hmax}$ orientations show a rather simple stress pattern without local perturbations. This is to a certain extent an expectable result since our model is in equilibrium with gravitation. Therefore neither isostatic effects as described


by e.g. Kaiser et al. (2005), Bada et al. (2001) or Jarosiński et al. (2006) nor local perturbations due to faults or fault zones (Kaiser et al., 2005; Jarosiński et al., 2006) can be considered since such perturbations are not implemented. Nevertheless, our model results also show no impact of mechanical contrasts on the orientation of $S_{Hmax}$, e.g predicted by Grünthal and Stromeyer (1994), Marotta et al. (2002) or Cacace (2008) despite mechanical contrast, e.g. a Young's modulus difference of 40 GPa between the sedimentary (30 GPa) and the upper crust units (70 GPa). Probably a lateral stiffness difference and a


weak unit seems to be necessary to get some perturbation due to a stiffness contrast (Reiter, 2020). However, our stiffness contrast between the sedimentary and upper crustal unit is vertical. To test this thesis, we defined an unrealistic low Young's modulus of 30 GPa to the upper crust unit of Avalonia (Fig. 5). In this case we could see perturbations at the border between the upper crust units of Avalonia and the EEC.

Although our kinematic boundary conditions applied are not derived from plate tectonic forces they fit in general the tectonic


setting of the model area. A shortening in N-S direction can be related to the alpine orogeny in the south and an extension in E-W direction correlates with the evolution of several extensional structures like the Cenozoic Rift System or the Eger Graben since the Paleogene (Kley et al., 2008).

### 5.2 Magnitudes of $S_{hmin}$, $S_{Hmax}$ and $S_V$

The $S_{hmin}$ magnitudes (Fig. 8a) in general show a very good correlation with the data of Morawietz et al. (2020) with a mean


difference of 3.3 MPa, a median difference of 0 MPa and an almost even distribution independent of data quality and depth. However, the model was calibrated with these values, so the almost perfect match should not be overrated.

The comparison of the $S_{Hmax}$ magnitudes does not show such a good match with a mean difference of 20.6 MPa and a median difference of 19.3 MPa (Fig. 9). Due to the calibration process described a much better fit should be achieved. But we have decided not to force a median of 0 MPa for various reasons. Compared to the $S_{hmin}$ magnitude data, the scattering is


significantly larger and the distribution is less even. This is an expectable result since $S_{Hmax}$ values are usually calculated and not measured and therefore $S_{Hmax}$ magnitudes have a lower reliability compared to $S_{hmin}$ values (Morawietz et al., 2020) but additionally there seems to be a dependency on depth and quality. A major part of the data indicating too large $S_{Hmax}$

magnitudes are from shallow depths (200 to 1000 m) (Fig. 9a). It can be assumed that the median and mean difference would be significantly better for a uniform depth distribution of the $S_{Hmax}$ values since the results below 1000 m show a good

match. A reduction of the $S_{Hmax}$ magnitudes in our model and thus a statistically better fit would therefore only lead to a better fit of the model result in the upper part of the model. Whether there is a dependency of the results on quality is difficult to assess, although data of quality B and C tend to show larger deviations than data of quality A. But most quality B and C data are also from shallow depths. Therefore, the depth dependency may overlay the quality dependency.

The stress magnitudes of the three hypothetical wells displayed in Fig. 10 show by and large the expected results. Since $S_V$ is

only dependent on the density the gradients of well 1 and 2 located in a single unit are constant all over the total well depth. The gradient of well 3 changes in between 1500 and 3500 m depth due to the change of units. The transition zone is quite large because of the vertical element resolution of about 800 m. Based on the sum of the overburden, the maximum $S_V$ at 10 km depth is the highest in well 1, followed by well 3 and 2. The stress differences between $S_{Hmax}$ and $S_{hmin}$ are dependent on the elastic rock properties. Therefore, these results again are mainly based on the unit the well is located in. Since the

Poisson's ratio is constant for the units involved (0.25) the Young's modulus is probably the decisive parameter. This explains the constant horizontal stress differences within well 1 and 2 and the variations over the length of well 3. In addition, the maximum stress differences seemed to be mainly dependent on the Young's modulus. Well 2 shows the smallest stress differences of 15 MPa and well 1 shows differences of 35 MPa according to a Young's Modulus of 30 and 70 GPa, respectively. An exception is well 2 with the highest total stress differences of up to 40 MPa.

Also the RSR displayed in Fig. 10 and 11 indicate a strong dependency on the Young's modulus since the highest values occur usually in the units of the upper crust characterized by a Young's modulus of 70 GPa and the lowest values are visible in areas with a high sediment thicknesses with a Young's modulus of 30 GPa. This correlation is almost perfectly visible in 4000 m depth in Fig. 11 in comparison to the depth of the top of the crystalline basement shown in Fig. 4. This stiffness difference between these two units is also responsible for the constant RSR in well 3 in the transition zone between these

units in 1500 to 3500 m depth (Fig. 10). The explanation for this correlation between the RSR and the Young's modulus are larger horizontal stresses due to a higher Young's modulus, which lead to a more compressive regime and vice versa. In addition, our results indicate a change of the stress regime with depth for the whole model area from a dominant strike-slip regime (1 < RSR < 2) to a normal faulting regime (RSR < 1) (Fig. 10 and 11). This change occurs, with few exceptions, between 2000 and 4000 m depth. The stress regime and thus in particular a change with depth is a decisive factor e.g. for the

wellbore stability, especially in case of directional or deviated drilling (Rajabi et al., 2016) or the stimulation of enhanced geothermal reservoirs (Azzola et al., 2019). Such depth dependent stress regimes are for example described by Brooke-Barnett et al. (2015), Cornet et al. (2007), Rajabi et al. (2016) and Rajabi et al. (2017).

To get a more detailed insight we compare our hypothetic wells 1, 2 and 3 with local magnitude data (Fig. 10). The model results of the hypothetic well 1 are displayed in comparison to values of the 'Kontinentale Tiefbohrung' (KTB), a major

scientific drilling project in Germany (Brudy et al., 1997). Our results of $S_V$ are in a very good agreement with the $S_V$ calculated from a mean density value. Only at greater depths the difference increases to about 5 MPa and in the uppermost 750 m our results are too large, possibly due to free surface effects caused by our model resolution. The results of $S_{hmin}$ and $S_{Hmax}$ show significantly larger differences to the data of Brudy et al. (1997). Except for the values at 3000 m depth the $S_{hmin}$ magnitudes of Brudy et al. (1997) are at least 15 MPa larger than the model results. The maximum difference of about 35

MPa is at 6400 m depth. The results of $S_{Hmax}$ show even greater deviations. All values of Brudy et al. (1997) are at least 15 MPa larger than the model results. The maximum difference is about 180 MPa at 7800 m depth and thus larger than our model results with 160 MPa. Remarkable is the change in the horizontal stress magnitudes of Brudy et al. (1997) at 3000 m depth. The $S_{hmin}$ magnitudes increase from 50 to 70 MPa within 200 m and $S_{Hmax}$ increases even by 30 MPa from 100 to 130 MPa. At the same time the inaccuracies also increase significantly. This can be attributed to the fact that the values between

3000 and 7000 m depth are only calculated and not directly measured (Brudy et al., 1997), which is why the values only got

a quality of worse than C (Morawietz et al., 2020) and are not used by us for calibration. A remarkable difference between our model and the geomechanical properties at the KTB site are the values of the Young's modulus. The calculated values of Brudy et al. (1997) are about 90 GPa on average between 3000 and 8000 m depth, which is about 20 MPa larger than our model assumption of 70 GPa in this area. Furthermore, in our model a normal faulting regime is established from about 3500 m depth downwards (Fig. 11), which is contrary to the stress regime of Brudy et al. (1997) showing a strike-slip regime from 1 km depth downwards. This indicate that our $S_{Hmax}$ are possibly too low within this model area. In general, our model results show a constant differential stress between $S_{hmin}$ and $S_{Hmax}$ of 35 MPa whereas the data of the KTB indicate an increasing differential stress with depth. The fact that our model does not include faults can also have an effect. The KTB is located above the Franconian Line and even intersecting it (Wagner et al., 1997). The Franconian Line is a major fault zone at the south-western margin of the Bohemian Massif with a polyphaser development from late Paleozoic to Neogene times (Zulauf, 1993; Peterek et al., 1997). In the end it is probably a combination of the lower Young's modulus in the model, the very large uncertainties of the calculated values and too low $S_{Hmax}$ values in our model, which may explain the differences of up to 180 MPa for the $S_{Hmax}$ magnitudes.

The stress magnitudes of the hypothetic well 2 are shown in comparison with $S_V$ and $S_{hmin}$ data from the NGB of Röckel and Lempp (2003). The $S_V$ values are in good agreement with our results down to ~2000 m depth, whereas the difference increases below this level. This shows that the density chosen for the sedimentary unit is at least appropriate for the upper part of this unit. A larger density would lead to better results at greater depths but since our calibration data mainly comes from depths shallower than 3500 m (Fig. 8 and 9), we consider assuming a density of 2300 kg m$^{-3}$ for the sedimentary unit is reasonable. The $S_{hmin}$ values indicate a good fit with our results across the entire depth range to 7000 m, which agrees with our general comparison shown in Fig. 8. Due to missing data, a comparison is not possible for the $S_{Hmax}$ values. However, Röckel and Lempp (2003) mention that the actual stress regime in the NGB can be characterized as normal faulting for the sub-salt level. At an average depth of the salt layer in the NGB of 4 km (Scheck-Wenderoth and Lamarche, 2005) our results show a normal faulting regime beneath 4 km depth too (Fig. 10 and 11).

For the comparison of our hypothetic well 3 we use data from the geothermal project in Soultz-sous-Forêts and some values from Rittershofen, another geothermal project nearby both located at the western edge of the URG. The URG is part of the major European Cenozoic Rift System located in south-western Germany and eastern France (Ziegler and Dèzes, 2006). The dashed $S_V$ gradients in Fig. 10 are calculated on the base of density values of Rittershofen (Azzola et al., 2019) correlated to the stratigraphic column of Soultz-sous-Forêts based on Aichholzer et al. (2016) up to 5080 m, the total depth of the deepest well in Soultz-sous-Forêts. Despite the density change between the sedimentary unit and the crystalline basement, the $S_V$ magnitudes of the model are in good agreement. An exception are the upper 750 m, where modelled $S_V$ magnitudes are slightly too low. The data of Valley and Evans (2007) show measured $S_{hmin}$ magnitudes between 1500 and 4500 m depth. They are in very good agreement with our model results within the upper 3000 m. In between 3500 and 4500 m depth the agreement is slightly worse but with a maximum difference of about 10 MPa at 4500 m depth still good. For the validation of the $S_{Hmax}$ magnitudes we use four calculated values of Klee and Rummel (1993) between 2200 and 3500 m. Within this depth interval our model results show a good correlation with a maximum deviation of about 5 MPa. Due to the small amount of data available for $S_{Hmax}$, a comparison with some calculated stress gradients can be helpful. Assumptions for stress gradients of Heinemann (1994), Klee and Rummel (1993) and Valley and Evans (2007) result in $S_{Hmax}$ magnitudes of 35 up to 55 MPa at a depth of 2000 m, of 95 up to 143 MPa at 5000 m and of 195 up to 310 MPa at 10 km. Even though these values show a quite wide range, the comparison of the model results allows the conclusion that our $S_{Hmax}$ values show a quite good agreement in the upper 5000 m but tend to be too low with increasing depth. This is also supported by the observation of seismic events which show a slight dominance of strike-slip versus normal faulting focal mechanisms at depths of 8 to 10 km in the URG (Cornet et al., 2007). In contrast, our results show a normal faulting regime, which implies to0 low $S_{Hmax}$ values.

In general, the model results show a good agreement with real magnitude data. The $S_V$ magnitudes show a good correlation for all three described cases of the KTB, the NGB and the Soultz-sous-Forêts site (Fig. 10). The modelled $S_V$ magnitudes appear to be only slightly too low with increasing depth but in general, the densities seem to be quite well chosen. A better agreement would be probably only possible with a higher stratigraphic resolution in the sedimentary unit and a density gradient within the upper crust. Such a simple gradient, which could also reduce the differences in the sedimentary unit, would be in this case less useful, because densities between two sedimentary units can differ considerably, independent of their depth. Overall, the $S_{hmin}$ magnitudes show a very good correlation. This can be seen in the general location-independent comparison in Fig. 8, but also with regard to the Soultz-sous-Forêts and NGB location in Fig. 10. Only the results for the KTB site show some considerable differences, with the greatest deviations from calculated $S_{hmin}$ values (3000 to 7000 m depth) that have not been directly measured and are of rather questionable quality. The $S_{Hmax}$ magnitudes show the largest deviations, both in the general comparison (Fig. 9) and in the local comparisons (Fig. 10). The general comparison shows that our values in the upper part are rather too large and at greater depths rather too small. The results of $S_{Hmax}$ at the KTB and Soultz-sous-Forêts sites only confirm the trend of too low values in the deeper parts, but not the trend of too large values in the shallower parts. In contrast, the $S_{Hmax}$ magnitudes at Soultz-sous-Forêts show a good agreement down to 2000 m depth and the values at the KTB site even indicate too low magnitudes down to 800 m depth. An indication for generally too small values with increasing depth are also the RSR values at 10 km depth (Fig. 10) which show larger areas of values lower than 0.5 indicating a radial extension, an uncommon stress regime in the upper crust. In the upper part of the model up to 4000 m the rather uniform tectonic regime, between normal faulting and transtensional, corresponds mainly to the tectonic conditions expected (Röckel and Lempp, 2003; Cornet et al., 2007). However, in detail, e.g. for the KTB site, the model cannot reflect differing local conditions. This could simply be a consequence of the simplifications made, which cannot resolve all local conditions, e.g. differing rock properties or nearby faults.

## 6. Conclusions

The model presented is the first 3D geomechanical model for Germany predicting the first order 3D stress tensor. The model is calibrated with $S_{Hmax}$ orientations from the WSM database and compilation of $S_{hmin}$ and $S_{Hmax}$ stress magnitude data from Morawietz et al. (2020). Overall, our model shows good results regarding the orientation of $S_{Hmax}$ and $S_{hmin}$ magnitudes despite the necessary simplifications due to the model resolution and rock property distributions as well as the highly irregular spread of the calibration data and their varying quality. The $S_{Hmax}$ orientations of the model are to a large extend within the uncertainty of the mean $S_{Hmax}$ orientations that are derived from the A to C quality data of the WSM database. Furthermore, the $S_{hmin}$ magnitudes show a quite good fit to various datasets (Röckel and Lempp, 2003; Valley and Evans, 2007), but the $S_{Hmax}$ magnitudes result show locally significant differences. Modelled $S_{Hmax}$ magnitudes are too small in the lower part of the model, whereas some results indicate too high values in the upper part. But in general, our model describes the regional 3D contemporary stress state quite well. Some larger deviations due to local structures are expectable. Therefore, the model results cannot be used for stress prediction on a local or reservoir scale as the resolution is not sufficient, but it can deliver initial stress conditions for smaller scale models that contain little or no stress magnitude data at all.

To improve our large-scale model a better stratigraphic resolution of the sedimentary unit and thus a better representation of the lithologies has to be implemented. This would increase the reliability of the comparison between measured stress magnitude data and the modelled ones. In addition to a vertical refinement, resolving lateral variations of the rock properties would be useful as well as these potentially account for lateral variability of the stress tensor.

**Data availability:**

The results of our model have been published and are publicly available (see Ahlers et al., 2021).

**Author contribution:**

Conceptualization of the project was done by AH, TH, KR, OH and BM. Construction and discretization of the model was done by SA. Data for the model and its calibration were collected and provided by SA, TH, LR, SM, MS and DA. Evaluation of the model results and their interpretation were performed by SA with the support of AH, TH, KR, BM, LR, OH and SM. SA wrote the paper with help from all coauthors. All authors read and approved the final paper.

**Competing interests:**

The authors declare that they have no conflict of interest.

**Acknowledgments:**

This study is part of the SpannEnD Project (www.SpannEnD-Projekt.de) which is supported by Federal Ministry for Economic Affairs and Energy (BMWI) and managed by Projektträger Karlsruhe (PTKA) (project code: 02E11637A). Coastlines and borders used in the figures are based on the Global Self-consistent Hierarchical High-resolution Geography (GSHHG) of Wessel and Smith (1996).

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
