# Peer review of "3D crustal stress state of Germany according to a data-calibrated geomechanical model"

_Solid Earth, 2020_

## Referee Comment (RC1) · Anonymous Referee #1 · 1 Feb 2021

A) General comments:

The manuscript by Ahlers et al., (se-2020-199) aims to evaluate the state of stress in Western Central Europe using a 3D geomechanical approach. Knowledge of contemporary stresses play a critical role in numerous practical applications and, therefore, the results of this paper/model can be used in different aspects of geomechanics in the study area. In particular, this large scale model can provide information on boundary conditions of any small scale models in Western Central Europe for any geothermal exploration/production and waste disposal.

Overall, the modelling technique used in this work (even the 'cookie-cutting' approach)

[Figure]

has been tested in several tectonic setting so far (I'm not sure why it has not been mentioned in this paper). This contribution is a valuable step forward in our collective efforts to stress modelling of the Western Central Europe. This is a very useful paper dealing with a topic which might be of broad interest to the readership of the Solid Earth.

While this study is a step forward modelling effort, some discussion is needed about what could be added in the next iteration to better understand the controls of the stress field in Western Central Europe. In particular, it is important to highlight how these "cookie-cutting" stress modelling could help us to evaluate the causes of tectonic stress in this region. I know the evaluation of the tectonic forces are not the scope of this paper, however, I think some discussion is required, as tectonic forces play the key role on the state of stress (note that these forces have been investigated, somehow, in the literature).

The geology, tectonic setting and the description of previous models (generally, the literature review part of the paper) needs more work (please see my specific comments below). In particular, for those who are not familiar with the tectonic setting of the region. I'd say readers need to know what really control the stresses in Western Central Europe and how this model can be used to deal with large tectonic forces? Again, I know the evaluation of tectonic forces are not the aim of this model, but some information would be really helpful for future work. If this model is a predictive one, then you need to provide some implication of the results. For example, how the stress changes (orientation of magnitude) predicted by this model, can be used in practical applications? All in all, this paper provide the first attempt on the regional scale 3D stress modelling in Western Central Europe, which provide interesting and useful information on the crustal stress of this region. Therefore, I'd suggest the publication of this paper following a moderate revision.

B) Specific comments:

B1) Title:

The current title says 'first results'. So, it is not clear if there will second/third results as future papers? I'd suggest to make this paper as a stand-alone one. The current title is a kind of confusing in a way that the second results will be different from this one. If yes, then someone might say what is the point of this paper if different results are going to be published sooner or later? So, some clarification would be appreciated.

B2) Abstract:

I am not a big fan of putting references in the Abstract. So, I'd suggest to remove the references and re-write the abstract to represent your work. I think the abstract needs to be re-written. By saying this "The model is open for further refinements regarding model geometry . . ." the readers might get confused as you are publishing and incomplete model and results. So, I'd suggest to be clear if it is a final model or not?

B3) Introduction:

I'd suggest to expand the introduction and provide some background on the stress forces and how stresses can be perturbed, at different scales (add some real examples as references). Here, you can also explain stress orders, and their importance (note that you have already talked about them at different sections, but never explained them). In the following section (i.e. previous models), you can explain how these orders of stress can affect state of stress in your area (based on the literature). In the beginning of the introduction, heaps of the cited papers are not included in the reference list! So, I am not sure if these papers are relevant to the statement or not (please see my comments in technical corrections for more details). So, I'd suggest to make sure that the cited references are appropriate and represent your statement.

B4) Fundamentals and state of the art:

The heading says 'state of the art'. So, some information on the 'state of the art' aspects of the work would be more than welcome.

The geology of the study area needs more information on the tectonic setting as well. I think a map is required to show the location of study area in relation to tectonic plate. This map and information then will help you to discuss if pull and push forces around your model can be explained by tectonic forces or not?

Figure 1 needs more work. In different parts of the paper, you mentioned different country names. Could you please add these names on the map? In addition, you need to add the dimension of the model (length of each edges) on this figure as well.

B5) Previous models:

I assume this part is a literature review section for the previous attempts on the stress modelling of the region. However, I believe, it is not well-organised and needs more detailed information. By reading this section, the readers need to understand the controls on the stress pattern in your study area. In the current version, you have explained that the previous results highlighted 'lateral stiffness contrasts in the lithosphere' and 'isostatic effects' are the main cause of 'stress perturbation'. But you did not discuss the causes of stress in the region? I am talking about the main forces that control the stress pattern (not those that provide perturbations).

Table 1: What is the difference between X and (X)?

B6) Model Setup:

The modelling setup and strategy has been widely used by the authors in a wide range of setting and scales. I'd suggest to add a statement in the beginning of Chapter 3 (i.e. modelling setup), and highlight that this setup and strategy has been used in both local and large scales (and cite them). In addition, in this section or somewhere else you need to explain that this model doesn't aim to evaluate the tectonic forces that control the stresses in this region. But, it mainly provides some information on the 3D description of stresses.

B7) Initial Stress State:
This section is really important and needs to be re-written. The current version is not clear. So, I'd suggest to give a better explanation on this paragraph.

B8) Displacement boundary conditions:

This section needs a bit of work as well. It needs to be clearer. So, I'd suggest to re-write this section.

Line 245: It's been mentioned" The orientations of the model boundaries are chosen parallel or perpendicular to the observed SHmax orientation". Could you please show it in a map? You probably can show the statistical results (that show the mean SH orientation) on Fig 2a? The current version of map in Fig 2a does not show it clearly, as there is not much data for the top right edge as well as whole western side of the model (and even the bottom edge!). So, it is not clear on what basis you are claiming that the model edges are parallel or perpendicular to SH.

Line 246-248: To me it is not clear how you have chosen to pull or push the model edges? So, it would be great to give us a better explanation on how these pull and pushing approach resemble the tectonic forces?

B9) Results:

Line 259: Why you did not calibrate the model with point-wise SH orientation? So, does it mean that you are calibrating your modelling results (in terms of SH orientation) with statistical models (inferred from WSM and pointwise data)? In all different part of the paper, it has been mentioned that the model will be calibrated against the WSM database for SH orientation. But, when we reach to the calibration stage, we see that the model is really calibrated with another statistical model (inferred from point-wise stress data). It is Okay, and I have no issue with the calibration procedure, but make it clear in the early part of the manuscript in order to avoid confusion.

Figure 6: I'd suggest to show WSM data on panel c as well.

Figure 9: Please use another colour for one of SH or SV. It is a bit difficult to follow

orange and red (quiet similar) in the plots. In addition, could you please show the different layers of the model on this figure? I also suggest to show the azimuth of SH orientation to see if SH rotate with depth, in particular once we move from one layer to another (basically change in layers means change in rock mechanical properties).

Figure 10: This is an interesting figure that clearly shows the variation of stress regime (represented by (RSR) with depth in your study area. However, I'd love to see the SH orientation on each of them as well. By showing the SH on each panel, you can clearly show/discuss if there is any variation for SH with depth or not.

B10) Discussion:

Change of predicted stress regime with depth is very important that needs more explanation. So, I expect more discussion on the implications of these changes at 1km, 2km and 4km. I expect to see how these changes can affect the geomechanics of sub-surface for any geothermal activity or waste disposal? As I mentioned somewhere else, I'd like to see the SH for each depths on each of these panels.

In addition, it would be great to provide some explanation on how these stress magnitude (and changes in RSR) evolve? Have you seen any similar changes in other regions with similar spatial scales (by means of stress data or stress models)? If yes, then would be great to cite them for those who are interested in this issue. Similar explanation is required for SH orientation. How does SH orientation evolve?

Line 377: Instead of saying 'good' I'd suggest to be quantitative. Of course you have explained it in the next sentences. But, I'd suggest to re-write these sentences to be more quantitative.

Line 382: You need some references where you have mentioned 'salt can act as mechanical contrast'.

Lines 391 and 392: I'd suggest to cite some papers who has explained the stress variations by using real data, not those who show modelling results.

Line 395: low far-field horizontal stresses? What does it mean? Some clarification would be great.

Line 405: The authors mentioned "our model results also show no impact of mechanical contrasts on the orientation of SHmax". So, I think some explanation here is required. In particular, it should be expanded in relation to Reiter 2020, where these parameters play a critical role on stress perturbation. So, why we do not see these orations here?

Line 409: Instead of saying 'very good' I'd suggest to be quantitative.

C) Technical corrections:

Lines 9 and 10: What do you mean by 'basic research likewise'? I'd suggest to be more specific.

Line 17: I'd suggest not to use "lithostratigraphic units" for such a large and regional scale model. It then could be confusing with the terminology used by the International Commission on Stratigraphy (https://stratigraphy.org/guide/litho). Similar issue in Lines 42 and 43.

Line 27: Bell (2003) & Kristiansen (2004) is not in the reference list! Please make sure that these papers have something replate to wellbore stability.

Lines 29 & 30: Altmann et al., (2014); Henk (2009); Smart et al., (2014); Hettema, (2020); Konstantinovskaya et al., (2012); Brady & Brown, (2004) are not in the reference list!

Line 30: use 'and' instead of 'or'

Line 31: Diederichs et al., (2004) is not in the reference list

Line 40: remove 'in this study'

Lines 41-43: These lines are repeated from abstract.

Line 49: Change it to Geology and Tectonic Setting of the study area.

Figure 2: I think it is modified from the one presented in Heidbach et al., (2018). So, it needs a reference! Do you think you really need to show this figure at all?

Lines 117 & 118: Put this statement "If several model versions are published by one author, the most current one is listed" in the Table caption.

Line 118: Remove "with a wide range"

Line 168: Be consistent. Sometimes five units, sometimes seven units.

Line 194: Please re-write this statement (In all the models used and also . . ... rocks.), as it is not clear.

Line 244: What do you mean by 'these' in this statement "At the five lateral boundaries of the model displacement boundary conditions are applied perpendicular to these"?

Line 332: Do not

Line 513: Conclusions

Line 546: The references list needs to be completed. There are many references that used in the text, but are not in the reference list.

---

## Referee Comment (RC2) · Anonymous Referee #2 · 15 Feb 2021

General

Ahlers et al. present a 3D geomechanical finite element model of Germany and surrounding that has been partly calibrated with observations. They compare calculated crustal stresses with available observations from the World Stress Map and other databases. Overall, the fit is acceptable and allows further analysis and model improvements.

The model seems to be the first step toward a complex model for Germany that can be used for dedicated local and regional stress field modelling. This is of major importance, as Germany restarted the search for a nuclear waste disposal facility, hence this

model and any successors will be much appreciated and exceptionally helpful.

The modelling approach is well established and has been used by some of the co-authors since many years. It is a combination of the stress models of (mainly) the Karlsruhe group with subsurface models of the GFZ group. The model development as presented in this manuscript was therefore, as I see it, a very easy task (mesh the sub-surface structures with Hypermesh - "push" with calibrated values in Abaqus - done). As no time-dependent material is involved, even computation time was likely short, de-spite more than 1 million elements. Additionally, as observations were partly used as constraints (model boundaries designed after stress orientation, stress values), it is no surprise that the fit of the model to the few available observations is at a good level. The model is thus just a start - but at the same time something that has been missed, which is the most important point on the positive side for this manuscript.

Frankly, I had problems to give my full support for this manuscript as (a) it is only a first and expected result and (b) the values (e.g. as grids in different depth slices in 2 km steps or so) are not even made available for the interested reader on GFZ websites or services like Pangaea.de. Furthermore, the presentation of the study and results can be partly improved, see my many suggestions below. A discussion of your results with previous models, independent if they are simple or not, should also be made.

I would emphatically ask the authors to revise the manuscript along my suggestions and consider publishing results like a gridded 3D stress field over depth, so that the manuscript deserves the role as stand-alone paper.

Specific comments

Title: the title is an example for smart exaggeration, but should be changed. The model and manuscript are part of a German project. You even write in lines 163ff "This area was chosen ... to simplify the definition of boundary conditions later on and with regard to important crustal structures which may affect the recent stress field in Germany... Additionally, model boundaries are selected distal to the German border

to avoid possible boundary effects in the area of main interest." So Germany is your goal, please use it in the title. Western Central Europe would also include Switzerland as whole, which is not completely part of your model. Also, avoid a phrase like "first results". There won't be a paper with "second results" and even "third results". How about "A data-calibrated geomechanical model for Germany - insights on the 3D crustal stress state".

Abstract: Please avoid citing references. You may use the term Western Central Europe here, but please emphasize that your area of interest is Germany, especially in light of the ongoing nuclear waste disposal location search. Add 2-3 more sentences on results, e.g. on the "salt problem".

L31f: This sentence needs references, e.g. from the search in Switzerland, Sweden and Finland. Add a short introduction on the ongoing search in Germany and why a 3D stress state model of Germany is desired.

L41, first 3D model for Germany: please add a few sentences with references to 3D geomechanical models for other parts of the world.

Section 2.1: should be expanded and more references should be added, e.g., regarding the evolution of the area. Kley & Voigt (2008) will be of help here.

L53/54: Please add reference for this crustal thickness statement (e.g. the works by Gregersen and colleagues or Mazur and colleagues).

L55, Tornquist-Teisseyre Zone and the Thor Suture: this needs more discussion, also in the setup of your model. Looking at Fig. 1b, the Trans European Suture Zone (TESZ) is shown only with the TTZ in Poland and the northern TESZ branch, the STZ in Denmark. In Fig. 1c you show the TESZ with TTZ in Poland (as above) but the more southerly located Thor Suture (southern branch) in Denmark. Nothing is said about the Tornquist Fan in between and the thickness variations here (see the works by Gregersen and colleagues, among others). Which line do you follow in Poland?

[Figure]

Mazur et al. (2015, Tectonics, https://doi.org/10.1002/2015TC003934) placed it a bit further southwest than commonly done before.

Fig. 1a: Please add a scale of 200 km (representing the search radius you apply later). Change the word "Location" to something more feasible.

Fig. 1b&c: Please update with a more appropriate representation of the whole TESZ. Which TESZ structure is included in the model, the one after Kley & Voigt or the one after Kroner et al.?

Fig. 1: Please increase font size of lat/long numbers.

Section 2.4: I miss a couple of sentences with presentation and discussion of the 3D models shown in Goes et al. (2000, GPC, https://doi.org/10.1016/S0921-8181(01)00057-1) and Warners-Ruckstuhl et al. (2013, GJI, https://doi.org/10.1093/gji/ggt219). These two as well as some in Table 1 should be picked up in the Discussion.

Section 3.1: You should add that you also neglect any remaining rebound effects due to the previous glaciation, see e.g., Brandes et al. (2015, Geology, https://doi.org/10.1130/G36710.1).

L143ff: How is the stress orientation calculated here? Do you use stress2grid for some grid points along a line and then calculate a mean? Which search radius is applied? Or just one coordinate representative for the center point of a boundary? What error can result for the stress orientation?

Section 3.2: You state Germany is your area of interest, and the model has some extension in west-east direction giving you some buffer around Germany. However, in the north-south direction your area of interest is very close to Germany's borders, much closer than in east-west direction. Why? Can any geometry effects be exluded in that direction?

Fig. 4: Avoid rainbow color scale! There are severeal scientific color scales nowadays

available (http://www.fabiocrameri.ch/colourmaps.php). Add that most of the caption information belongs to the subfigures in the lower left. Country borders should also be found in the main subfigures, otherwise it is awkward for the reader to retrieve suitable information from the figure.

L205f: How reliable is the assumption of vertical boundaries here?

Fig. 5: The ALCAPA section in the upper right appears to be much smaller in geographical extent than the one shown in Fig. 1c, where the whole southern part is covered with ALCAPA. Is the line in Fig. 1c misplaced? Did you change your model geometry?

L254f: Although clearly without significance for the result, did you consider to make your model roughly some few 100 metres bigger & smaller in size (as you can roughly pre-calcuclate those extensions and shortenings) so that you get ∼correct coordinates "today"?

Section 4.1., first paragraph: This description irritates the reader. You talk about two grids where you compare nearest grid points? Why don't you calculate the database values on the centre points of the elements in the FE model? Or calculate values from two sources on an identical grid?

L264: Please add reference for this statement.

L265: Please add reference for the <25°. Why not 15° or 22.5°?

L267: Why 5 km?

Fig. 6: I suggest to add two more subfigures. It would be interesting to have a subfigure depicting the number of stress data in each grid point of the WSM grid and one which shows the standard deviation of each grid point. It might help to compare if outliers in the histogram (d) or the map (b) fall together with those.

Fig. 6d indicates that your model needs a stress orientation change of roughly 10°.

Have you tested that?

Figs. 7&8: Suggest to (i) split the quality color in (a) into two each for above and below 1 or 1.5 km, and (ii) color-code the histogram in (c) with the 6 colors so that one can distinguish, especially in Fig. 8c, the different quality and depth sources.

Fig. 9: Please make profile lines thicker. I also suggest to add the RSR over depth to this figure (or create separate figure). In view of this figure, I suggest to calculate the misfit (weighted sum of squared model minus observation difference divided by observation error) for each profile and quantity, and use them in the discussion. Can be even used in future studies with improved models.

Fig. 10: Please use different color scale (not rainbow). Scale should be 0 to 3. I suggest to check earthquake catalogues if focal mechanisms are available for some of these depth slices and plot them too. There are some remarkable edge effects in the upper 4000 m, or shall it be a true thrust mechanism?

Discussion: Please compare your results also to previous models listed in Table 1, but also the values shown in Figures 4 & 10 of Warners-Ruckstuhl et al. (2013).

L400: yet?

L452ff: Here it would be good to have RSR over depth in Fig. 9.

L488ff: Here it would be nice to refer to focal mechanisms in Fig. 10.

Technical comments:

L110: Move '(Fig. 1a)' after 'database' as otherwise the reader thinks your model is shown in Fig. 1a.

L142: initial stress

L259f: Sentence sounds awkward, suggest: 'A mean SHmax orientation is used on a regular 0.5° grid, as we do not use individual data records for this comparison.'
[Figure]

L332: do not

L437: comma after '(KTB)'

---

## Author Comment (AC1) · 9 Mar 2021

Answer to anonymous referee #1:

A) General comments: The manuscript by Ahlers et al., (se-2020-199) aims to evaluate the state of stress in Western Central Europe using a 3D geomechanical approach. Knowledge of contemporary stresses play a critical role in numerous practical applications and, therefore, the results of this paper/model can be used in different aspects of geomechanics in the study area. In particular, this large scale model can provide information on boundary conditions of any small scale models in Western Central Europe for any geothermal exploration/production and waste disposal. Overall, the modelling

technique used in this work (even the 'cookie-cutting' approach) has been tested in several tectonic setting so far (I'm not sure why it has not been mentioned in this paper). This contribution is a valuable step forward in our collective efforts to stress modelling of the Western Central Europe. This is a very useful paper dealing with a topic which might be of broad interest to the readership of the Solid Earth. While this study is a step forward modelling effort, some discussion is needed about what could be added in the next iteration to better understand the controls of the stress field in Western Central Europe. In particular, it is important to highlight how these "cookie-cutting" stress modelling could help us to evaluate the causes of tectonic stress in this region. I know the evaluation of the tectonic forces are not the scope of this paper, however, I think some discussion is required, as tectonic forces play the key role on the state of stress (note that these forces have been investigated, somehow, in the literature). The geology, tectonic setting and the description of previous models (generally, the literature review part of the paper) needs more work (please see my specific comments below). In particular, for those who are not familiar with the tectonic setting of the region. I'd say readers need to know what really control the stresses in Western Central Europe and how this model can be used to deal with large tectonic forces? Again, I know the evaluation of tectonic forces are not the aim of this model, but some information would be really helpful for future work. If this model is a predictive one, then you need to provide some implication of the results. For example, how the stress changes (orientation of magnitude) predicted by this model, can be used in practical applications? All in all, this paper provide the first attempt on the regional scale 3D stress modelling in Western Central Europe, which provide interesting and useful information on the crustal stress of this region. Therefore, I'd suggest the publication of this paper following a moderate revision.

We thank the anonymous referee #1 for reviewing our manuscript carefully and for his/her suggestions. In the following we will address the specific comments in detail.

B) Specific comments: B1) Title: The current title says 'first results'. So, it is not clear

if there will second/third results as future papers? I'd suggest to make this paper as a stand-alone one. The current title is a kind of confusing in a way that the second results will be different from this one. If yes, then someone might say what is the point of this paper if different results are going to be published sooner or later? So, some clarification would be appreciated.

We have changed the title to '3D crustal stress state of Germany according to a data-calibrated geomechanical model'

B2) Abstract: I am not a big fan of putting references in the Abstract. So, I'd suggest to remove the references and re-write the abstract to represent your work. I think the abstract needs to be re-written. By saying this "The model is open for further refinements regarding model geometry ..." the readers might get confused as you are publishing and incomplete model and results. So, I'd suggest to be clear if it is a final model or not?

We have removed the references. Since we have removed '- first results' from the title, we think there should be no misunderstanding regarding the last sentence anymore.

B3) Introduction: I'd suggest to expand the introduction and provide some background on the stress forces and how stresses can be perturbed, at different scales (add some real examples as references). Here, you can also explain stress orders, and their importance (note that you have already talked about them at different sections, but never explained them). In the following section (i.e. previous models), you can explain how these orders of stress can affect state of stress in your area (based on the literature).

We have added a short introduction regarding different orders of stress sources in Sect. 2.2.

In the beginning of the introduction, heaps of the cited papers are not included in the reference list! So, I am not sure if these papers are relevant to the statement or not (please see my comments in technical corrections for more details). So, I'd suggest to

make sure that the cited references are appropriate and represent your statement.

We apologize for the missing citations throughout the introduction, there was a problem linking them to our reference list, but this has been corrected.

B4) Fundamentals and state of the art: The heading says 'state of the art'. So, some information on the 'state of the art' aspects of the work would be more than welcome. The geology of the study area needs more information on the tectonic setting as well. I think a map is required to show the location of study area in relation to tectonic plate. This map and information then will help you to discuss if pull and push forces around your model can be explained by tectonic forces or not?

We have added some additional information (line 70ff and 75ff) about the evolution from cretaceous to recent times including potential stress sources which we take up again in the discussion (line 454ff). We think that a map showing the tectonic plates forces is not useful here since this could give the reader the impression that we assume this as boundary conditions for the model.

Figure 1 needs more work. In different parts of the paper, you mentioned different country names. Could you please add these names on the map? In addition, you need to add the dimension of the model (length of each edges) on this figure as well.

We have added a new subfigure (1 b) with country names and the model dimensions. B5) Previous models:

I assume this part is a literature review section for the previous attempts on the stress modelling of the region. However, I believe, it is not well-organised and needs more detailed information. By reading this section, the readers need to understand the controls on the stress pattern in your study area. In the current version, you have explained that the previous results highlighted 'lateral stiffness contrasts in the lithosphere' and 'isostatic effects' are the main cause of 'stress perturbation'. But you did not discuss the causes of stress in the region? I am talking about the main forces that control the

stress pattern (not those that provide perturbations).

We think that a detailed literature review regarding the stress sources is not required here since the objective of our study is not to identify the stress sources.

Table 1: What is the difference between X and (X)?

You are right, we forgot to mention it. The (X) describes the same model approach as we use, where the boundary conditions are not derived from the plate boundary forces, but still represent them. We have added a statement to the table caption (line 141f).

B6) Model Setup: The modelling setup and strategy has been widely used by the authors in a wide range of setting and scales. I'd suggest to add a statement in the beginning of Chapter 3 (i.e. modelling setup), and highlight that this setup and strategy has been used in both local and large scales (and cite them). In addition, in this section or somewhere else you need to explain that this model doesn't aim to evaluate the tectonic forces that control the stresses in this region. But, it mainly provides some information on the 3D description of stresses.

Thank you for your suggestion, we have added a statement at the end of the introduction (line 45ff) and we have and we have clarified that this model does not aim to evaluate the sources of stress (line 150ff).

B7) Initial Stress State: This section is really important and needs to be re-written. The current version is not clear. So, I'd suggest to give a better explanation on this paragraph.

You are right this is an important section. We have rewritten the whole section and added a new figure (line 251-272).

B8) Displacement boundary conditions: This section needs a bit of work as well. It needs to be clearer. So, I'd suggest to re-write this section. Line 245: It's been mentioned" The orientations of the model boundaries are chosen parallel or perpendicular to the observed SHmax orientation". Could you please show it in a map? You probably

can show the statistical results (that show the mean SH orientation) on Fig 2a? The current version of map in Fig 2a does not show it clearly, as there is not much data for the top right edge as well as whole western side of the model (and even the bottom edge!). So, it is not clear on what basis you are claiming that the model edges are parallel or perpendicular to SH.

We have added the mean SHmax orientations to the new subfigure (1 b).

Line 246-248: To me it is not clear how you have chosen to pull or push the model edges? So, it would be great to give us a better explanation on how these pull and pushing approach resemble the tectonic forces?

We have rewritten this paragraph (line 275ff). Additionally, please see our comment to B6).

B9) Results: Line 259: Why you did not calibrate the model with point-wise SH orientation? So, does it mean that you are calibrating your modelling results (in terms of SH orientation) with statistical models (inferred from WSM and pointwise data)? In all different part of the paper, it has been mentioned that the model will be calibrated against the WSM database for SH orientation. But, when we reach to the calibration stage, we see that the model is really calibrated with another statistical model (inferred from point-wise stress data). It is Okay, and I have no issue with the calibration procedure, but make it clear in the early part of the manuscript in order to avoid confusion.

The reasons why we use a mean SHmax orientation is described in Sect 4.1. We now have made it clear in the entire manuscript (e.g. line 159, line 281).

Figure 6: I'd suggest to show WSM data on panel c as well.

We think that this information is not helpful in this figure. The detailed comparison is now shown Fig. 1a and b.

Figure 9: Please use another colour for one of SH or SV. It is a bit difficult to follow orange and red (quiet similar) in the plots. In addition, could you please show the

different layers of the model on this figure? I also suggest to show the azimuth of SH orientation to see if SH rotate with depth, in particular once we move from one layer to another (basically change in layers means change in rock mechanical properties).

You are right the lines have been quite thin and the colors were quite similar. We have changed the color of SHmax and increased the thicknesses of the lines. This is a good suggestion! We added the model units at the right side of the plots. We think that is it not necessary to show the SHmax orientations in this figure since they are constant over the entire depth shown. However, we mention this in line 419f now and added a corresponding statement to the figure caption in line 365f.

Figure 10: This is an interesting figure that clearly shows the variation of stress regime (represented by (RSR) with depth in your study area. However, I'd love to see the SH orientation on each of them as well. By showing the SH on each panel, you can clearly show/discuss if there is any variation for SH with depth or not.

Please see our comment to Fig. 9.

B10) Discussion: Change of predicted stress regime with depth is very important that needs more explanation. So, I expect more discussion on the implications of these changes at 1km, 2km and 4km. I expect to see how these changes can affect the geomechanics of sub-surface for any geothermal activity or waste disposal? As I mentioned somewhere else, I'd like to see the SH for each depths on each of these panels. In addition, it would be great to provide some explanation on how these stress magnitude (and changes in RSR) evolve? Have you seen any similar changes in other regions with similar spatial scales (by means of stress data or stress models)? If yes, then would be great to cite them for those who are interested in this issue. Similar explanation is required for SH orientation. How does SH orientation evolve?

You are right this is an important point. We have expanded the discussion section regarding the stress regime change with depth and added some references (line 485 - 497).

Line 377: Instead of saying 'good' I'd suggest to be quantitative. Of course you have explained it in the next sentences. But, I'd suggest to re-write these sentences to be more quantitative.

We have restructured these two sentences (line 416ff).

Line 382: You need some references where you have mentioned 'salt can act as mechanical contrast'.

You are right, we have added some references in line 423f.

Lines 391 and 392: I'd suggest to cite some papers who has explained the stress variations by using real data, not those who show modelling results.

You are right, it is better to cite some papers with 'real data' here. We have removed the 'modelling papers' and added some additional references with 'real data' in line 427f.

Line 395: low far-field horizontal stresses? What does it mean? Some clarification would be great.

To make it clearer, we now additionally use the term 'high order stress source'. Please see also our comment to B3)

Line 405: The authors mentioned "our model results also show no impact of mechanical contrasts on the orientation of SHmax". So, I think some explanation here is required. In particular, it should be expanded in relation to Reiter 2020, where these parameters play a critical role on stress perturbation. So, why we do not see these orientations here?

You are right with regard to Reiter (2020) this needs a little more explanation. We have added a small discussion on this topic and calculated an additional model to verify our hypothesis (line 446- 452).

Line 409: Instead of saying 'very good' I'd suggest to be quantitative.

We have rewritten this statement (line 459f).

C) Technical corrections: Lines 9 and 10: What do you mean by 'basic research like-wise'? I'd suggest to be more specific.

We think it is enough to mention the specific topics in the introduction (line 25-32).

Line 17: I'd suggest not to use "lithostratigraphic units" for such a large and regional scale model. It then could be confusing with the terminology used by the International Commission on Stratigraphy (https://stratigraphy.org/guide/litho). Similar issue in Lines 42 and 43.

To avoid misunderstandings, we have changed it to 'sedimentary unit' (e.g. in line 17)

Line 27: Bell (2003) & Kristiansen (2004) is not in the reference list! Please make sure that these papers have something replate to wellbore stability.

You are right, please see our comment to B3)

Lines 29 & 30: Altmann et al., (2014); Henk (2009); Smart et al., (2014); Hettema, (2020); Konstantinovskaya et al., (2012); Brady & Brown, (2004) are not in the refer-ence list!

You are right, please see our comment to B3)

Line 30: use 'and' instead of 'or'

We have changed it, as suggested

Line 31: Diederichs et al., (2004) is not in the reference list

Please see our comment to B3)

Line 40: remove 'in this study'

We have removed it

Lines 41-43: These lines are repeated from abstract.

Yes, they are almost similar since it is a very important statement.

Line 49: Change it to Geology and Tectonic Setting of the study area.

Thank you for the suggestion, we have changed it.

Figure 2: I think it is modified from the one presented in Heidbach et al., (2018). So, it needs a reference! Do you think you really need to show this figure at all?

We have added the reference in line 104. Yes, we think this small figure is very useful for readers who are not so familiar with this topic and the definition of the reduced stress tensor.

Lines 117 & 118: Put this statement "If several model versions are published by one author, the most current one is listed" in the Table caption.

You are right it is good the mention it again in the table captions. We have added it (line 134f).

Line 118: Remove "with a wide range"

We have removed it.

Line 168: Be consistent. Sometimes five units, sometimes seven units.

You are right, we have changed it.

Line 194: Please re-write this statement (In all the models used and also ..... rocks.), as it is not clear.

We are sorry but we cannot understand where this sentence is incomprehensible to you. The top of the basement is not necessarily the top of the crystalline rock. However, usually the top of the basement is also defined as the base of the sedimentary layer. But, e.g. low-grade metamorphic rocks are more like sedimentary than crystalline rocks

Line 244: What do you mean by 'these' in this statement "At the five lateral boundaries of the model displacement boundary conditions are applied perpendicular to these"?

You are right this was not clear we have rewritten this paragraph.

Line 332: Do not

Thank you, we have changed it as mentioned.

Line 513: Conclusions

We have changed it, as suggested.

Line 546: The references list needs to be completed. There are many references that used in the text, but are not in the reference list.

Please see our comment to B3).

---

## Author Comment (AC2) · 9 Mar 2021

Answer to anonymous referee #2:

Ahlers et al. present a 3D geomechanical finite element model of Germany and surrounding that has been partly calibrated with observations. They compare calculated crustal stresses with available observations from the World Stress Map and other databases. Overall, the fit is acceptable and allows further analysis and model improvements. The model seems to be the first step toward a complex model for Germany that can be used for dedicated local and regional stress field modelling. This is of major importance, as Germany restarted the search for a nuclear waste disposal

facility, hence this model and any successors will be much appreciated and exceptionally helpful. The modelling approach is well established and has been used by some of the coauthors since many years. It is a combination of the stress models of (mainly) the Karlsruhe group with subsurface models of the GFZ group. The model development as presented in this manuscript was therefore, as I see it, a very easy task (mesh the subsurface structures with Hypermesh - "push" with calibrated values in Abaqus - done).As no time-dependent material is involved, even computation time was likely short, despite more than 1 million elements. Additionally, as observations were partly used as constraints (model boundaries designed after stress orientation, stress values), it is no surprise that the fit of the model to the few available observations is at a good level. The model is thus just a start - but at the same time something that has been missed, which is the most important point on the positive side for this manuscript. Frankly, I had problems to give my full support for this manuscript as (a) it is only a first and expected result and (b) the values (e.g. as grids in different depth slices in 2 km steps or so) are not even made available for the interested reader on GFZ websites or services like Pangaea.de. Furthermore, the presentation of the study and results can be partly improved, see my many suggestions below. A discussion of your results with previous models, independent if they are simple or not, should also be made. I would emphatically ask the authors to revise the manuscript along my suggestions and consider publishing results like a gridded 3D stress field over depth, so that the manuscript deserves the role as stand-alone paper.

We thank the anonymous referee #2 for reviewing our manuscript carefully and for his/her suggestions. The complete modeling results are now published and available under: https://doi.org/10.48328/tudatalib-437 In the following, we will address the specific comments in detail.

Specific comments Title: the title is an example for smart exaggeration, but should be changed. The model and manuscript are part of a German project. You even write in lines 163ff "This area was chosen ... to simplify the definition of boundary conditions

later on and with regard to important crustal structures which may affect the recent stress field in Germany... Additionally, model boundaries are selected distal to the German border to avoid possible boundary effects in the area of main interest." So Germany is your goal, please use it in the title. Western Central Europe would also include Switzerland as whole, which is not completely part of your model. Also, avoid a phrase like "first results". There won't be a paper with "second results" and even "third results". How about "A data-calibrated geomechanical model for Germany - insights on the 3D crustal stress state".

Thank you for your suggestion, we changed the title to '3D crustal stress state of Germany according to a data-calibrated geomechanical model'.

Abstract: Please avoid citing references. You may use the term Western Central Europe here, but please emphasize that your area of interest is Germany, especially in light of the ongoing nuclear waste disposal location search. Add 2-3 more sentences on results, e.g. on the "salt problem".

We removed the references and avoid the term 'Western Central Europe' now. However, we did not add more sentences on the results. We think that a brief overview of our results is sufficient for this abstract. If we would address, for example the 'salt problem' in the abstract we would also need to address several other specific issues, which may influence our results. Instead, the salt topic, among others, is treated in the discussion section.

L31f: This sentence needs references, e.g. from the search in Switzerland, Sweden and Finland. Add a short introduction on the ongoing search in Germany and why a 3D stress state model of Germany is desired.

We have added some references, e.g. for Germany and Switzerland (line 30f) but no further details since the focus of this study is not on the search for a nuclear waste repository in Germany and we don't think that a more detailed description is necessary here.

L41, first 3D model for Germany: please add a few sentences with references to 3D geomechanical models for other parts of the world.

We have added some references to other 3D geomechanical models at the end of the introduction in line 45f.

Section 2.1: should be expanded and more references should be added, e.g., regarding the evolution of the area. Kley & Voigt (2008) will be of help here.

We have expanded Sect. 2.1. regarding the evolution from the cretaceous to recent times. (line 70-77).

L53/54: Please add reference for this crustal thickness statement (e.g. the works by Gregersen and colleagues or Mazur and colleagues).

We have added a reference to Mazur et at., 2015 (line 54).

L55, Tornquist-Teisseyre Zone and the Thor Suture: this needs more discussion, also in the setup of your model. Looking at Fig. 1b, the Trans European Suture Zone (TESZ) is shown only with the TTZ in Poland and the northern TESZ branch, the STZ in Denmark. In Fig. 1c you show the TESZ with TTZ in Poland (as above) but the more southerly located Thor Suture (southern branch) in Denmark. Nothing is said about the Tornquist Fan in between and the thickness variations here (see the works by Gregersen and colleagues, among others). Which line do you follow in Poland? Mazur et al. (2015, Tectonics, https://doi.org/10.1002/2015TC003934) placed it a bit further southwest than commonly done before.

This may be a misunderstanding since in the first version, we did not use the terms TTZ and STZ completely correct. In figure 1d (former 1c) we show the crustal units of Western Central Europe. To avoid further misunderstandings we have removed the term TTZ and use the term Tornquist Suture now (line 55 and figure 1d). The Tornquist Suture together with the Thor Suture in the northwest represents the border between the EEC and Avalonia, based on Kroner et al. (2010). In figure 1 c (former b) we show

the younger tectonic framework based on Kley and Voigt (2008) which includes the TTZ and STZ. In our model only the Thor and Tornquist sutures are implemented as boundary between the crustal units of Baltica and the EEC as displayed in Fig. 5. The tectonic framework in figure 5 is only indirectly represented in the model by the varying thickness of the sedimentary unit (see Fig. 4).

Fig. 1a: Please add a scale of 200 km (representing the search radius you apply later). Change the word "Location" to something more feasible.

Since we are using a Mercator projection for the plot, it would not be correct to add a scale. However, we have added the dimensions of the model edges in the new figure 1b. This should be sufficient for orientation. We have changed 'location' to 'location of measurement'.

Fig. 1b&c: Please update with a more appropriate representation of the whole TESZ. Which TESZ structure is included in the model, the one after Kley & Voigt or the one after Kroner et al.?

Please see our comments above. We have expanded Sect. 2.1 (line 70-77) and edited Fig 1c and d.

Fig. 1: Please increase font size of lat/long numbers.

You are right the font size was quite small. We have increased it.

Section 2.4: I miss a couple of sentences with presentation and discussion of the 3D models shown in Goes et al. (2000, GPC, https://doi.org/10.1016/S0921-8181(01)00057-1) and Warners-Ruckstuhl et al. (2013, GJI, https://doi.org/10.1093/gji/ggt219). These two as well as some in Table 1 should be picked up in the Discussion.

You are right, the publication of Warners-Ruckstuhl et al., 2013 has been missing in our table. We have added it. We did not add the publication of Goes et al, 2000 since there are no results shown regarding the crustal stress field in our model area and the

stress model mentioned of Loohuis et al, 2001 (in preparation) seems not to have been published.

Section 3.1: You should add that you also neglect any remaining rebound effects due to the previous glaciation, see e.g., Brandes et al. (2015, Geology, https://doi.org/10.1130/G36710.1).

This is already mentioned, e.g. in the discussion (line 443ff).

L143ff: How is the stress orientation calculated here? Do you use stress2grid for some grid points along a line and then calculate a mean? Which search radius is applied? Or just one coordinate representative for the center point of a boundary? What error can result for the stress orientation?

The calculation is described in detail in Sect. 4.1. However, we have replaced 'observed' with 'mean' to avoid further misunderstandings and added a reference to the mean SHmax orientations displayed in the new subfigure (line 178).

Section 3.2: You state Germany is your area of interest, and the model has some extension in west-east direction giving you some buffer around Germany. However, in the north-south direction your area of interest is very close to Germany's borders, much closer than in east-west direction. Why? Can any geometry effects be exluded in that direction?

As mentioned in Sect. 3.2 (line 172ff) the model area was chosen with regard to (1) the orientation of SHmax (2) important crustal structures which may affect the recent stress field in Germany e.g. the Bohemian Massif, the Avalonia-EEC boundary and the European Cenozoic Rift System. and (3) additionally distal to the German border. Therefore, the focus was not the distance to the German border. If this had been the only criterion, the model area could have been chosen much smaller, like in the north. Geometry effects cannot be excluded but as shown in Fig. 11 they occur outside of Germany, also in the north.

[Figure]

Fig. 4: Avoid rainbow color scale! There are several scientific color scales nowadays available (http://www.fabiocrameri.ch/colourmaps.php). Add that most of the caption information belongs to the subfigures in the lower left. Country borders should also be found in the main subfigures, otherwise it is awkward for the reader to retrieve suitable information from the figure.

Thank you for your suggestion! We have changed the color bars and added country borders to the depth maps. As the subfigures are quite small, it is almost impossible to add all information to these figures. Instead, we have added an enlarged and labeled figure of the database of the top of crystalline basement as a supplement.

L205f: How reliable is the assumption of vertical boundaries here?

This is difficult to assess, but due to the very poor data situation, this seemed to be the most sensible choice. There are no arguments supporting other geometries.

Fig. 5: The ALCAPA section in the upper right appears to be much smaller in geographical extent than the one shown in Fig. 1c, where the whole southern part is covered with ALCAPA. Is the line in Fig. 1c misplaced? Did you change your model geometry?

You are right, the boundary of the ALCAPA unit was wrong. The displayed boundary was the alpine deformation front and not the boundary used in the model. We have corrected it with information of Brückl et al., (2010).

L254f: Although clearly without significance for the result, did you consider to make your model roughly some few 100 metres bigger & smaller in size (as you can roughly pre-calcuclate those extensions and shortenings) so that you get correct coordinates "today"?

As you mentioned, this will not matter for our model size. If this is required for smaller or higher resolution models, there is the possibility to rectify the model geometry afterwards so that the results can be captured at the correct 'today' coordinates.

Section 4.1., first paragraph: This description irritates the reader. You talk about two grids where you compare nearest grid points? Why don't you calculate the database values on the centre points of the elements in the FE model? Or calculate values from two sources on an identical grid?

We have used a grid of 0.5°, as we find this suitable for a model of our resolution. For comparison we have to interpolate the results on a plane first since the nodes of our mesh are located at different depths. In the end we use the nearest values to avoid additional uncertainties due to a second interpolation.

L264: Please add reference for this statement.

This is a definition made in stress2grid.

L265: Please add reference for the <25. Why not 15 or 22.5?

This is the maximum standard deviation of the WSM data used (WSM quality A to C) for the calculation of the mean SHmax orientation. To make it easier to understand, especially since we now mention the standard deviation of the derived grid too, we have now replaced the SD with the WSM quality (line 290f).

L267: Why 5 km?

Since the SHmax orientations do not show significant changes with depth we use a medium depth, where we did not expect any effects of the topography and which is well validated by our calibration data.

Fig. 6: I suggest to add two more subfigures. It would be interesting to have a subfigure depicting the number of stress data in each grid point of the WSM grid and one which shows the standard deviation of each grid point. It might help to compare if outliers in the histogram (d) or the map (b) fall together with those.

We think that the standard deviation and/or the numbers of data used for each grid is no information that creates significant added value and for which two further figures

are not necessary. However, we mention the median of the standard deviation in the text now (line 297). Anyone interested in this specific information can easily calculate it from the WSM data using the stress2grid tool. All settings used are mentioned in the text (line 290ff).

Fig. 6d indicates that your model needs a stress orientation change of roughly 10 Have you tested that?

Yes, we have tested it, but we were not able to get a better result.

Figs. 7&8: Suggest to (i) split the quality color in (a) into two each for above and below 1 or 1.5 km, and (ii) color-code the histogram in (c) with the 6 colors so that one can distinguish, especially in Fig. 8c, the different quality and depth sources.

That is a good suggestion! We have color-coded the histograms depending on the quality, but we think that it is not useful to add three additional colors.

Fig. 9: Please make profile lines thicker. I also suggest to add the RSR over depth to this figure (or create separate figure). In view of this figure, I suggest to calculate the misfit (weighted sum of squared model minus observation difference divided by observation error) for each profile and quantity, and use them in the discussion. Can be even used in future studies with improved models.

Yes, the lines have been quite thin. We have increased the thickness of the lines and added the RSR over depth. We are not able to quantify our model error, so we cannot calculate such a misfit. However, you are right, stress model uncertainties are an interesting and important subject.

Fig. 10: Please use different color scale (not rainbow). Scale should be 0 to 3. I suggest to check earthquake catalogues if focal mechanisms are available for some of these depth slices and plot them too. There are some remarkable edge effects in the upper 4000 m, or shall it be a true thrust mechanism?

We have changed the colors and the scale. Yes, these are edge effects we mention it in

the figure caption now (line 412f). In general, it is a good idea to consider focal mechanisms (FMS) as an additional data base for comparison with model results. However, we decided not to show FMS in comparison to our model results. Although FMS can represent the stress state of the crust, this is not always the case. It is because a focal mechanism is a kinematic information at first hand. If the fault is not optimally oriented in the contemporary stress fields or if there is slip partioning among several faults, slip on a single fault may not represent the stress state, e.g. in the San Andreas fault system there are pure strike-slip earthquakes on some faults whereas there are parallel fault strands where slip exhibits a strong thrust-component to accommodate the overall slight oblique relative motion between the Pacific and North America Plate. The same is true on the North Anatolian Fault in the Sea of Marmara where most earthquakes on the main fault are pure strike-slip events despite the extension going on, which is taken up by subsidiary faults. In addition, depth information on hypocenters is generally poorly constrained. Things are different when not single FMS are compared to the stress field but inversion of FSM is performed, since a presumption in the inversion is that all earthquakes occurred in the same homogenous stress field. However, due to low seismicity rates in most parts of Germany not many inversions have been done and moreover an inversion result is representative for the area in which the earthquakes have occurred which would allow comparison only for larger areas. Furthermore, the model shows that the regime changes with depth, so that the depth of the FMS plays a central role in a comparison. However, the focal depth is subject to large uncertainties, especially for the low magnitude earthquakes, so that the focal depth is often set at 10 km. The true depth position is often not known.

Discussion: Please compare your results also to previous models listed in Table 1, but also the values shown in Figures 4 & 10 of Warners-Ruckstuhl et al. (2013).

We already mention almost all models listed in Table 1 in Sect 5.1. We have not added a detailed discussion in comparison to the results of Warners-Ruckstuhl et al., 2013 since the results do not show a specific cause of stress perturbation regarding our

model area.

L400: yet?

No, we are not planning to implement it.

L452ff: Here it would be good to have RSR over depth in Fig. 9.

Please see our comment above.

L488ff: Here it would be nice to refer to focal mechanisms in Fig. 10.

Please see our comment above.

Technical comments: L110: Move '(Fig. 1a)' after 'database' as otherwise the reader thinks your model is shown in Fig. 1a.

Thank you, we have changed it, as suggested.

L142: initial stress

Thank you, we have changed it, as suggested.

L259f: Sentence sounds awkward, suggest: 'A mean SHmax orientation is used on a regular 0.5 grid, as we do not use individual data records for this comparison.'

You are right, we have restructured the sentence.

L332: do not

Thank you, we have changed it, as suggested.

L437: comma after '(KTB)'

Thank you, we have changed it, as suggested.

---

## Author Response (AR2)

Answer to anonymous referee #2:

I am happy with the revised manuscript, which has improved a lot. I am pleased to recommend publication. Please deal with the small number of technical corrections in the process to follow.

L30 remove full stop
L150 this -> these
L234 Please add ALCAPA explanation for completeness.
L292 Fig -> Figs
L380 show -> shows
L386 shows thus -> thus shows
L393 Fig -> Figs
L420 orientation -> orientations
L485 & 493 Fig -> Figs
L516 indicate -> indicates
L528 & 533 Fig -> Figs
L552 to0 -> too the publication of this paper following a moderate revision.

We thank the anonymous referee #2 for his/her technical corrections, we have implemented them as suggested.

Table 2: is there a reference for 130 GPa of Young's modulus in the lithospheric mantle?

Table 2: Since the Young's modulus of the lithospheric mantle is only roughly estimated, there is no reference for this value.